# DeTikZify: Synthesizing Graphics Programs for Scientific Figures and Sketches with TikZ

**Jonas Belouadi**[*]    **Simone Paolo Ponzetto**[†]    **Steffen Eger**[‡]

Natural Language Learning Group[*,‡] Data and Web Science Group[†]
University of Mannheim[*,†] University of Technology Nuremberg[‡]
{jonas.belouadi,ponzetto}@uni-mannheim.de, steffen.eger@utn.de

## Abstract

Creating high-quality scientific figures can be time-consuming and challenging, even though sketching ideas on paper is relatively easy. Furthermore, recreating existing figures that are not stored in formats preserving semantic information is equally complex. To tackle this problem, we introduce DeTikZify, a novel multimodal language model that automatically synthesizes scientific figures as semantics-preserving TikZ graphics programs based on sketches and existing figures. To achieve this, we create three new datasets: DaTikZ$_{v2}$, the largest TikZ dataset to date, containing over 360k human-created TikZ graphics; SketchFig, a dataset that pairs hand-drawn sketches with their corresponding scientific figures; and MetaFig, a collection of diverse scientific figures and associated metadata. We train DeTikZify on MetaFig and DaTikZ$_{v2}$, along with synthetically generated sketches learned from SketchFig. We also introduce an MCTS-based inference algorithm that enables DeTikZify to iteratively refine its outputs without the need for additional training. Through both automatic and human evaluation, we demonstrate that DeTikZify outperforms commercial Claude 3 and GPT-4V in synthesizing TikZ programs, with the MCTS algorithm effectively boosting its performance. We make our code, models, and datasets publicly available.[1]

## 1 Introduction

Creating high-quality scientific figures is similar to typesetting scientific documents in many ways. When it comes to typesetting, markup languages like LaTeX enjoy widespread popularity, as exemplified by major machine learning conferences that either mandate or strongly encourage LaTeX-formatted submissions.[2] The advantages of using such languages go beyond producing high-quality outputs; documents expressed as high-level, semantics-preserving programs enhance accessibility, serve archival purposes, and remain easily editable and human-readable (facilitating language modeling applications; Moosavi et al., 2021; Lu et al., 2023). Consequently, efforts have been made to recover this type of information from outputs stored in lower-level vector graphics formats like PDF or SVG, or raster graphics formats (Desai et al., 2021; Blecher et al., 2024). At the other end of the spectrum, the versatility of LaTeX comes with a steep learning curve, and typesetting can often be challenging for end users. In response, researchers have been working on assisting authors with certain aspects of the problem, such as typesetting math based on hand-drawn sketches (Kirsch, 2010; Wu et al., 2020).

Just like documents, scientific figures can also be created using markup languages. A popular example is the TikZ graphics language (Tantau, 2023), which can be integrated into LaTeX documents, providing comparable benefits and encountering similar challenges. However, unlike LaTeX, the prospects of TikZ in research contexts remain largely unexplored. Although the promise of simplifying editing and

---

[1]https://github.com/potamides/DeTikZify
[2]https://{neurips.cc,icml.cc,iclr.cc}/Conferences/2024/CallForPapers

38th Conference on Neural Information Processing Systems (NeurIPS 2024).

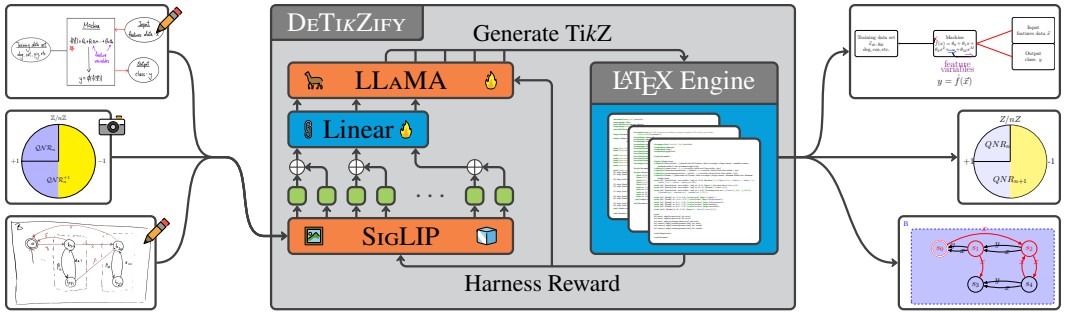

Figure 1: Overview of the DeTi*κ*Zify architecture: A multimodal language model converts sketches or figures into Ti*k*Z programs, which are compiled by a LᴬTᴇX engine. This provides a reward signal to the model via MCTS, allowing it to iteratively refine the output until satisfactory results are achieved.

enabling applications in visual understanding (Masry et al., 2022; Huang et al., 2023) is evident, there are currently no viable solutions for recovering graphics programs from compiled figures. Moreover, there is a lack of tools that assist in creating graphics programs, e.g., based on hand-drawn sketches, despite the clear demand for such approaches on the TᴇX Stack Exchange (TᴇX.SE),[3] where nearly 10% of all questions revolve around Ti*k*Z, making it the most frequently discussed topic on the site. Addressing this gap could greatly improve the accessibility of existing figures and support researchers at all levels of programming proficiency when creating new ones, fostering diversity and inclusion. In response, we introduce DeTi*κ*Zify, a multimodal language model that automatically synthesizes Ti*k*Z programs for scientific figures and sketches (cf. Figure 1). Our key contributions are as follows:

(i) As part of DeTi*κ*Zify, we introduce (a) DaTi*κ*Z_{v2}, a large Ti*k*Z dataset with over 360k human-created Ti*k*Z graphics; (b) SketchFig, a dataset of human-created sketches with paired scientific figures; and (c) MetaFig, a large meta-dataset of scientific figures and associated texts.

(ii) We train DeTi*κ*Zify on MetaFig and DaTi*κ*Z_{v2}, augmented with synthetic sketches that mimic SketchFig. We demonstrate that DeTi*κ*Zify can effectively synthesize Ti*k*Z programs for both existing scientific figures and sketches, outperforming the commercial large language models (LLMs) GPT-4V and Claude 3 (OpenAI, 2023b; Anthropic, 2024).

(iii) We also present an inference algorithm based on Monte Carlo Tree Search (MCTS) that is tailored to graphics programs and allows DeTi*κ*Zify to iteratively refine *its own outputs* for a given computational budget, further improving performance without additional training.

## 2   Related Work

**Image-to-LᴬTᴇX Conversion**   A closely related task is the translation of mathematical illustrations into LᴬTᴇX markup. In inspirational work, Kirsch (2010) tackle the recognition of single hand-drawn symbols to find corresponding LᴬTᴇX commands. Subsequent works by Deng et al. (2017); Zhang et al. (2017, 2019); Wu et al. (2020); Wang and Liu (2021) expand on this concept to handle hand-drawn and scanned math formulas. Suzuki et al. (2003); Wang and Liu (2020); Blecher et al. (2024); Lv et al. (2023) further extend the scope by extracting LᴬTᴇX formulas alongside text from entire documents.

**Image Vectorization**   Similarly, converting (rasterized) figures into Ti*k*Z programs can be characterized as a form of image vectorization (Sun et al., 2007; Diebel, 2008; Ganin et al., 2018; Li et al., 2020; Ma et al., 2022; Zhu et al., 2024). Most existing methods vectorize images into low-level graphics primitives in the SVG format (Tian and Günther, 2024). Although this works well for specific domains like fonts, icons, and emoji (Lopes et al., 2019; Carlier et al., 2020; Reddy, 2021; Rodriguez et al., 2023b), it does not capture higher-level semantics and does not generalize well to our scientific context (cf. Appendix B). Closer to our work, Ellis et al. (2018) generate vector representations as graphics programs based on a limited subset of LᴬTᴇX commands. Their approach even handles

---

[3]https://tex.stackexchange.com

sketches, but their experiments are restricted to a synthetic dataset with only basic shapes of limited complexity. Belouadi et al. (2024) also generate TikZ programs, but their primary emphasis is on conditioning the generation on textual descriptions, with images serving only as a secondary input.

**Code Generation** As TikZ is implemented in the Turing-complete TEX macro system (Erdweg and Ostermann, 2011), our work is also closely tied to code generation (Xu et al., 2022). Despite continuing progress in this field (Chen et al., 2021; Li et al., 2022, 2023; Guo et al., 2024; Lozhkov et al., 2024), most research concentrates on high-resource languages like Python, Java, and JavaScript (Zan et al., 2023), typically overlooking TEX in evaluations. However, TEX and TikZ may still find their way into the training data, as demonstrated by the zero-shot ability of some models to understand and generate code in these languages (Bubeck et al., 2023; Belouadi et al., 2024; Sharma et al., 2024).

## 3 Datasets

We introduce DaTikZ_v2, to our knowledge, the most comprehensive dataset of TikZ graphics to date; SketchFig, the first dataset comprising human-created sketches of scientific figures; and MetaFig, a large-scale scientific figure dataset with rich metadata. See Appendix E for examples.

**DaTikZ_v2** DaTikZ_v2 serves as the primary source of TikZ graphics for training DeTikZify. It is an expanded version of DaTikZ_v1 (Belouadi et al., 2024), incorporating graphics from the same sources, namely curated repositories, TEX.SE, arXiv papers, and artificial examples. The key difference is that DaTikZ_v2 includes all TikZ programs that compile

| Source | DaTikZ_v1 | DaTikZ_v2 |
|---|---|---|
| curated | 981 | 1 566 |
| TEX.SE | 29 238 | 30 609 |
| arXiv | 85 656 | 326 450 |
| artificial | 1 957 | 1 958 |
| all | 117 832 | 360 583 |

Table 1: Breakdown of the number of unique TikZ graphics in DaTikZ_v2 compared to its predecessor DaTikZ_v1.

with TEX Live 2023,[4] regardless of whether they have associated captions, which was a requirement for inclusion in DaTikZ_v1 but is not needed for DeTikZify. This approach allows us to create a dataset that is more than three times as large as its predecessor (cf. Table 1).

**SketchFig** To create realistic synthetic sketches of scientific figures in DaTikZ_v2, we rely on examples of real human-created sketches. TEX.SE is a suitable source for collecting these, as users often illustrate their questions with sketches, and the answers provide the desired figure. We semi-automatically extract these figure-sketch pairs by first ranking all questions on the site that contain images based on their similarity to the string "a sketch of a scientific figure" using a multimodal vision encoder (Zhai et al., 2023). We retain the ones with high similarity scores, manually filter for true positives, and align them with the best matching figure provided in the answers. In total, we collect 549 figure-sketch pairs this way. As we also want to use this dataset for evaluation (cf. §6), we ensure that for a subset of these sketches, no code provided in the answers is included in DaTikZ_v2.

**MetaFig** Beyond TikZ graphics, there is a much larger pool of figures where the underlying source is not available. Existing datasets that collect such figures frequently come with rich metadata, such as captions, OCR tokens, and paragraphs that mention the figures (Hsu et al., 2021; Karishma et al., 2023; Rodriguez et al., 2023a). Since such high-level descriptions are useful for pretraining (cf. §4; Liu et al., 2023b), we collect these datasets and merge them with the subset of figures in DaTikZ_v2 that have captions. This results in over 734k figure-text pairs, more than twice the size of DaTikZ_v2.

## 4 The DeTikZify Model

Building on previous work (Liu et al., 2023b,a; Dai et al., 2023; McKinzie et al., 2024), we build DeTikZify by combining a pretrained vision encoder with a pretrained language model (cf. Figure 1), where the vision encoder receives figures or sketches as input images, and the language model generates corresponding TikZ programs as output. We focus on code language models that have been pretrained on TEX, as this prior knowledge may be helpful for our task. All the models we end up using follow the LLaMA architecture (Touvron et al., 2023): CodeLLaMA (Rozière et al., 2023) has likely been trained on TEX code from arXiv (Touvron et al., 2023), as has been TinyLLaMA (Zhang et al.,

---

[4]https://tug.org/texlive

2024), while DEEPSEEK (code variant; Guo et al., 2024) was trained on TEX code from GitHub. For the vision encoder, we use SIGLIP (Zhai et al., 2023), which has been trained on OCR annotations (Chen et al., 2023c) and demonstrates state-of-the-art understanding of text-rich images (Tong et al., 2024; Chen et al., 2023b), a crucial skill for our task. We then condition the LLMs on SIGLIP's patch embedding vectors. To reduce the prompt length, we concatenate adjacent patch embeddings (Chen et al., 2023a). A feed-forward layer with dimensions $2\delta_{\text{SIGLIP}} \times \delta_{\text{LLM}}$ serves as a connector, mapping image features of dimension $\delta_{\text{SIGLIP}}$ to the LLM word embedding space of dimension $\delta_{\text{LLM}}$.

**Model Training** We experiment with TINYLLAMA$_{1.1\text{B}}$ and DEEPSEEK$_{1.3\text{B}}$ (approximately 1 billion parameters each) and CODELLAMA$_{7\text{B}}$ and DEEPSEEK$_{7\text{B}}$ (7 billion parameters each). When referring to specific variants of DETiKZIFY, we use the names DETiKZIFY-TL$_{1.1\text{B}}$, DETiKZIFY-DS$_{1.3\text{B}}$, DETiKZIFY-CL$_{7\text{B}}$, and DETiKZIFY-DS$_{7\text{B}}$, respectively. For all models, we use the SOVIT$_{400\text{M}}$ variant of SIGLIP as the vision encoder. Following Liu et al. (2023b,a), we first pretrain the connector with other model parameters frozen. We pretrain for one epoch on METAFIG with ADAMW (Loshchilov and Hutter, 2019), a batch size of 256, a learning rate of 1e−3, and a cosine learning rate decay with a 3% warmup ratio. Next, we unfreeze the language model (keeping the vision encoder frozen) and fine-tune on examples from DATiKZ$_{\text{v2}}$ that fit within a 2048 token context window. We use a batch size of 128, a learning rate of 4e−5, and train for three epochs. Training data ablations can be found in Appendix B.

**Synthetic Sketches** When training DETiKZIFY on DATiKZ$_{\text{v2}}$, we randomly replace figures with synthetic sketches 50% of the time. Sketches are generated on the fly, meaning that each time a figure is sampled as a sketch, a different synthetic sketch will be generated. Creating realistic sketches requires high-level image manipulation methods that go beyond traditional transformations like zooming or cropping. We, therefore, adopt INSTRUCT-PIX2PIX (Brooks et al., 2023), a model capable of diversely editing images based on human instructions. We chose this model due to its remarkable zero-shot performance in generating synthetic sketches during our initial experiments. By then fine-tuning the model on SKETCHFIG, we further improve its performance (cf. §7 and Appendix C).

## 5 Iterative Refinement with Monte Carlo Tree Search

Due to the inherent probabilistic nature of language models, generating valid TikZ programs during inference can be a challenging task. The generated code may not always comply with the syntactic and semantic rules of TEX and TikZ, potentially leading to compilation errors. While constrained decoding algorithms can assist in guiding models towards generating valid programs (Ugare et al., 2024; Poesia et al., 2022; Scholak et al., 2021), these approaches are limited to programming languages defined by context-free grammars (CFGs). However, TEX and TikZ are not defined by CFGs (Erdweg and Ostermann, 2011), rendering these methods ineffective for our purpose. Moreover, even if the generated code compiles successfully, fidelity errors such as misaligned elements, inconsistent scaling, repetitions, or mislabeling may only become apparent in the rendered output.

Despite these challenges, which make it difficult to guide DETiKZIFY based on intermediate states, we can still analyze completed outputs in a straightforward manner (e.g., by examining compiler diagnostics or comparing rendered outputs to the input image), allowing us to make informed decisions during subsequent sampling iterations. This concept of making decisions based on random sampling of the search space forms the core of Monte Carlo Tree Search (MCTS; Coulom, 2007). By integrating DETiKZIFY with MCTS and adapting the standard MCTS algorithm to our problem domain, we can iteratively steer DETiKZIFY towards more promising regions of the output space (cf. Figure 1). In the following, we outline our fundamental approach, with further extensions discussed in Appendix A.

### 5.1 Integrating MCTS into DETiKZIFY

MCTS is a versatile search algorithm that has been successfully applied to various domains, including board games (Silver et al., 2016, 2017), procedural content generation (Kartal et al., 2016a,b; Summerville et al., 2015), and more recently, guiding language models to achieve long-term goals (Brandfonbrener et al., 2024; Zhang et al., 2023b; Chaffin et al., 2022). The algorithm incrementally builds a search tree and repeatedly runs simulations until an exit condition is met or a computational budget is exhausted. In our context, at depth $n$, each node's state consists of $n$ lines of TikZ code, and edges represent continuations for generating the next line. Initially, MCTS starts with only an empty root node and then iteratively performs the following four steps (cf. Figure 2):

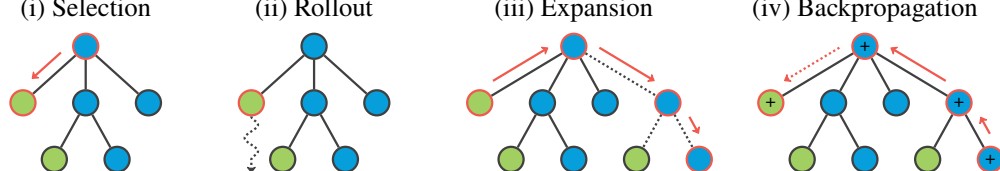

(i) Selection      (ii) Rollout      (iii) Expansion      (iv) Backpropagation

Figure 2: An example of the four steps of an MCTS simulation: The selection policy (i) reaches a green backtracking node (normal nodes are blue), causing new nodes from the rollout (ii) to be added to the parent node during expansion (iii). The reward is backpropagated (iv) accordingly.

**Selection**    Each simulation starts at the root node and successively selects child nodes based on a *selection policy* until a leaf node is reached. The policy determines which parts of the tree should be explored further, balancing the *exploitation* of high-value regions and *exploration* of less-visited areas. Following previous work, we use Upper Confidence Trees (UCT; Kocsis and Szepesvári, 2006) as our selection policy, iteratively selecting the successor node $i$ that maximizes the formula

$$\text{UCT}(i) = \frac{\sum_{j=1}^{n_i} V_{i,j}}{n_i} + c\sqrt{\frac{\ln(n_{\text{p}(i)})}{n_i}}, \tag{1}$$

where $V_{i,j} \in [-1, 1]$ is the estimated value of $i$ at the $j$th visit, $n_i$ and $n_{\text{p}(i)}$ are the visit counts at $i$ and its parent $\text{p}(i)$, respectively, and $c$ is a coefficient that controls the degree of exploration.

**Rollout**    Once a leaf node is selected, we utilize DETIKZIFY as a *rollout policy*. By conditioning it on the node's state, we continue to sample TikZ code until the end-of-sequence token is encountered. This so-called rollout is then stored for reuse in the subsequent steps.

**Expansion**    Next, the tree is *expanded* by adding nodes from the rollout as new leaf nodes. While most implementations add only one node (i.e., one line of TikZ code) per simulation, computing rollouts with LLMs is computationally expensive. Therefore, inspired by MCTS for real-time settings (Soemers et al., 2016), we instead add multiple nodes. Specifically, we add $\sqrt{|r| - d_l}$ new nodes, where $|r|$ is the number of lines in rollout $r$ and $d_l$ is the depth of the old leaf node $l$. This approach allows our tree to grow quickly in early simulations while converging to the standard case in the long run. To enable the tree to grow in multiple directions, we also introduce *backtracking* nodes (Brandfonbrener et al., 2024; Chaslot et al., 2008). For each added node $i$, we add a backtracking node as a sibling that mirrors the parent node $\text{p}(i)$. When a backtracking node is expanded, its descendants are added to $\text{p}(i)$ so that the backtracking node remains a leaf. This enables a practically infinite search space anywhere in the tree while still maintaining a bounded branching factor.

**Backpropagation**    Finally, we calculate the value for rollout $r$ using a predefined reward function (cf. §5.2) and *backpropagate* it to every node $i$ on the path from the root node to the newly added nodes by appending it to $V_{i,:}$. We also increment the visit counts $n_i$ for the same nodes. For backtracking nodes, only the visit counts are updated. Finally, we check any exit conditions. If MCTS terminates, we return the TikZ program of the rollout that achieved the highest value.

## 5.2    Reward Functions

We explore two distinct reward functions to guide the search process. The first reward function utilizes compiler diagnostics to identify documents that compile successfully. The second reward function provides a visual signal based on perceptual image similarity, which, in addition, helps find TikZ programs that better match the input image. We explore further reward functions in Appendix A.

**Compiler Diagnostics**    The diagnostics-based reward function is based on analyzing the log file from compiling the generated TikZ program. We assign rewards according to the error state and whether an output file was produced. The reward function is defined as follows:

$$V_{i,j} = \begin{cases} 1 & \text{if the code compiles without issues,} \\ 0 & \text{if the code compiles with recoverable errors,} \\ -1 & \text{if compilation fails due to a fatal error.} \end{cases} \tag{2}$$

| | Reference Figures | | | | | | | Synthetic Sketches | | | | | | |
|---|---|---|---|---|---|---|---|---|---|---|---|---|---|---|
| Models | MTE↑ | cBLEU↑ | TED↓ | DSim↑ | SSim↑ | KID↓ | AVG↑ | MTE↑ | cBLEU↑ | TED↓ | DSim↑ | SSim↑ | KID↓ | AVG↑ |
| Claude 3 | 51.812 | 0.111 | 57.389 | 64.896 | 83.372 | 17.822 | 0.148 | 50.156 | 0.024 | 59.731 | 59.102 | 73.954 | 29.541 | 0.189 |
| GPT-4V | 61.975 | 0.286 | 57.178 | 69.741 | 86.215 | 6.714 | 0.612 | 54.126 | 0.024 | 60.298 | 61.98 | 75.687 | 33.203 | 0.15 |
| DT-TL$_{1.1B}$ | 88.03 | 1.168 | 58.815 | 65.538 | 84.161 | 15.747 | 0.207 | 90.597 | 0.502 | 60.202 | 60.585 | 77.947 | 21.851 | 0.454 |
| DT-DS$_{1.3B}$ | 83.771 | 1.336 | 57.661 | 68.659 | 86.079 | 11.536 | 0.572 | 87.446 | 0.541 | 60.112 | 62.756 | 79.097 | 17.334 | 0.642 |
| DT-CL$_{7B}$ | 88.593 | 1.477 | 56.893 | 72.315 | 87.466 | 8.301 | 0.869 | 91.221 | 0.555 | 59.563 | 65.118 | 79.717 | 12.207 | 0.941 |
| DT-DS$_{7b}$ | 82.366 | 1.815 | 57.227 | 73.01 | 88.323 | 5.951 | 0.965 | 89.299 | 0.69 | 59.693 | 65.198 | 80.207 | 12.207 | 0.965 |

Table 2: System-level scores for output-driven inference (DeTi*κ*Zify abbreviated as DT). Bold and underlined values indicate the best and second-best scores for each metric column, respectively. Cell shading reflects the relative score magnitudes across input types. Arrows indicate metric directionality.

**Self-Assessed Perceptual Similarity (SelfSim)**   SelfSim computes the reward as the *perceptual similarity* (Zhang et al., 2018) between the input image and the compiled output figure. We hypothesize that DeTi*κ*Zify *itself* can assess this similarity, enabling the model to guide its own search process. To achieve this, we encode both images into embedding vectors using DeTi*κ*Zify's vision encoder and calculate SelfSim as their cosine similarity (Fu et al., 2023; Hessel et al., 2021). In cases where compilation fails, we assign a reward of -1. In §7, we demonstrate that SelfSim correlates well with human judgments and outperforms other baseline methods.

# 6   Experiments

Before training on DaTi*κ*Z$_{v2}$, we extract 1k samples to serve as our test set for an automatic evaluation and generate corresponding synthetic sketches. To mitigate data leakage from pretraining to testing, we only include items created after the cut-off date of CodeLLaMA and exclude repositories that may have been used in training DeepSeek. We also use an *n*-gram matching algorithm to prevent cross-contamination with our train split (OpenAI, 2023a). For a human evaluation involving human-created sketches, we also select 100 items from SketchFig that do not overlap with DaTi*κ*Z$_{v2}$ (cf. §3). Across all models, we set the temperature to 0.8 and the exploration coefficient $c$ to 0.6. We provide examples of real and synthetic sketches as well as generated outputs in Appendix E and Table 4.

**Baselines**   Given Claude 3 and GPT-4V's potential for our task (cf. §2), we use them as baselines. Similar to DeTi*κ*Zify, we instruct these models to generate Ti*k*Z programs for given images. However, as proprietary chatbots, they often mix code and natural language (Zhang et al., 2023c; Belouadi et al., 2024) and do not expose the internals needed to compute SelfSim. This makes it impractical to apply our MCTS-based refinement algorithm, which is designed for code-only outputs and open models. Instead, we compare our approach to equivalent chat-oriented refinement methods, i.e., we use Self-Refine as an alternative to diagnostics-based MCTS and Visual Self-Refine as an alternative to SelfSim-based MCTS (Madaan et al., 2023; cf. Appendix C for additional inference details). In Appendix B, we also explore SVG as an alternative to Ti*k*Z but find it less effective for our domain.

## 6.1   Automatic Evaluation

We introduce two inference tasks to automatically evaluate our models on the test split of DaTi*κ*Z$_{v2}$. During *output-driven* inference (OI), we employ the diagnostics-based reward and use successful compilation as an early exit condition (we consider compilation successful if an output artifact is produced). For *time-budgeted* inference (TI), we use the more fine-grained SelfSim-based reward and continue from OI until a computational budget of 10 minutes is exhausted (cf. Brandfonbrener et al., 2024), investigating the extent of achievable improvement. We report results for the two use cases where either (rasterized) reference figures or (synthetic) sketches serve as model inputs (cf. §1). Due to high inference costs, we only evaluate commercial Claude 3 and GPT-4V in OI using Self-Refine, leaving TI with Visual Self-Refine for human evaluation. We evaluate the following properties:

**Code Similarity**   To measure the similarity between generated and reference Ti*k*Z programs, we use CrystalBLEU (cBLEU), a variant of BLEU optimized for evaluating code (Eghbali and Pradel, 2023; Papineni et al., 2002), and the TEX Edit Distance (TED), our adapted version of the Extended Edit Distance (Stanchev et al., 2019) combined with a TEX tokenizer.

| Models | Reference Figures | | | | | | | Synthetic Sketches | | | | | | |
| --- | --- | --- | --- | --- | --- | --- | --- | --- | --- | --- | --- | --- | --- | --- |
| | MST↑ | cBLEU↑ | TED↓ | DSim↑ | SSim↑ | KID↓ | AVG↑ | MST↑ | cBLEU↑ | TED↓ | DSim↑ | SSim↑ | KID↓ | AVG↑ |
| DT-TL$_{1.1B}$ | **33.775** | −0.011 | −2.001 | +8.704 | +5.561 | −12.146 | 0.128 | **35.975** | +0.094 | −0.628 | +5.82 | +3.026 | +0.854 | 0.014 |
| DT-DS$_{1.3B}$ | 29.975 | −0.028 | −1.303 | +8.464 | +5.108 | **−8.728** | 0.531 | 32.429 | +0.061 | −0.504 | +5.573 | +2.685 | +5.493 | 0.22 |
| DT-CL$_{7B}$ | 25.124 | +0.07 | **−1.351** | **+7.797** | **+4.93** | −4.868 | **0.876** | 26.219 | +0.073 | −0.468 | **+5.079** | +2.455 | +5.493 | 0.681 |
| DT-DS$_{7b}$ | 24.145 | **−0.073** | −1.542 | +6.974 | +3.893 | −0.946 | 0.76 | 26.195 | +0.054 | **−0.696** | +4.887 | +2.241 | **+1.099** | **0.994** |

Table 3: System-level scores for time-budgeted inference, displaying relative changes for metrics shared with output-driven inference (Table 2; colored green for improvements and red for declines) and absolute scores for independent metrics. Bold and underlined values indicate the best and second-best *absolute* scores for each metric column, respectively. Arrows indicate metric directionality.

**Image Similarity**  In addition to SELFSIM (SSIM), which can also be used as a metric, we report DREAMSIM (DSIM; Fu et al., 2023), a fine-tuned metric for perceptual similarity. We also compute the Kernel Inception Distance (KID $\times 10^3$; Bińkowski et al., 2018), which assesses the overall quality of generated figures by comparing their distribution with the distribution of reference figures. These metrics are always computed by comparing the generated figures to the reference figures, regardless of what the model receives as input.

**Average Similarity**  To offer a holistic view of each model's performance, we also compute the arithmetic mean (AVG) of all code and image similarity metrics. Given that these metrics operate on different scales, we min-max normalize their scores before calculating the average.

**Efficiency**  For OI, we compute the Mean Token Efficiency (MTE) as the 10% winsorized mean of the ratio of the number of tokens in the final TikZ program to the total number of tokens generated to arrive at that program. For TI, we instead compute the Mean Sampling Throughput (MST), measuring the throughput of unique TikZ graphics for the given budget.

**Results**  Table 2 presents the system-level metric scores for OI. As expected, the scores for reference figures are, on average, 38% higher than those for synthetic sketches, but similar patterns emerge across both input types. DETikZIFY-CL$_{7B}$ and DETikZIFY-DS$_{7B}$ consistently outperform all other models, achieving AVG scores of 0.869 & 0.965 for figures and 0.941 & 0.965 for sketches, respectively. In contrast, GPT-4V reaches AVG scores of only 0.612 and 0.15, placing it in competition with the smaller 1b models: for figures, GPT-4V surpasses DETikZIFY-TL$_{1.1B}$ and DETikZIFY-DS$_{1.3B}$, which achieve scores of 0.207 and 0.572, respectively. However, these smaller models outperform GPT-4V on sketches, where they achieve scores of 0.454 and 0.642. CLAUDE 3 trails behind all our models, with an AVG of only 0.148 and 0.189. When examining individual similarity metrics, DETikZIFY-DS$_{7B}$, the top-performing DETikZIFY model overall, surpasses GPT-4V, the best baseline, by more than 3pp (percentage points) on average for DREAMSIM and SELFSIM, while maintaining a noticeably lower KID. In terms of cBLEU, GPT-4V, and CLAUDE 3 only reach 6.5–18.5% of the performance achieved by the lowest-scoring DETikZIFY model (DETikZIFY-TL$_{1.1B}$). The differences in TED are less pronounced, possibly due to the influence of boilerplate code, which cBLEU inherently ignores.

For efficiency, all DETikZIFY models demonstrate an MTE of 82–91%, indicating that only 1–2 out of 10 inference runs require a second simulation to generate a compilable TikZ program. Interestingly, the model size does not seem to particularly influence this score, with the pretraining setup appearing to be the key factor instead. For instance, DETikZIFY-TL$_{1.1B}$ and DETikZIFY-CL$_{7B}$ share a similar pretraining setup and exhibit comparable MTE values, as do DETikZIFY-DS$_{1.3B}$ and DETikZIFY-DS$_{7B}$. We can further observe that (i) MTE is generally higher for sketches compared to figures, and (ii) for figures, the MTE of similarly pretrained models is inversely correlated with their scores on other metrics. These phenomena likely stem from models making fewer mistakes when the input is less detailed or when their understanding of it is limited—a finding that aligns well with other studies (Tong et al., 2024). Compared to DETikZIFY, CLAUDE 3 and GPT-4V perform considerably worse, with an MTE of only 50–62%. Notably, for these models, 98.5% of the items already compile after the initial Self-Refine step, meaning that this inefficacy primarily originates from the natural language texts surrounding the code and that Self-Refine is nearly equivalent to regular sampling-based inference.

The results for DETikZIFY on TI are presented in Table 3. Remarkably, increasing the computational budget for MCTS improves nearly all metrics for both reference figures and sketches as input without requiring access to any additional knowledge. The improvement with sketches is particularly noteworthy, as it demonstrates that the refinement process enhances the desired properties even when

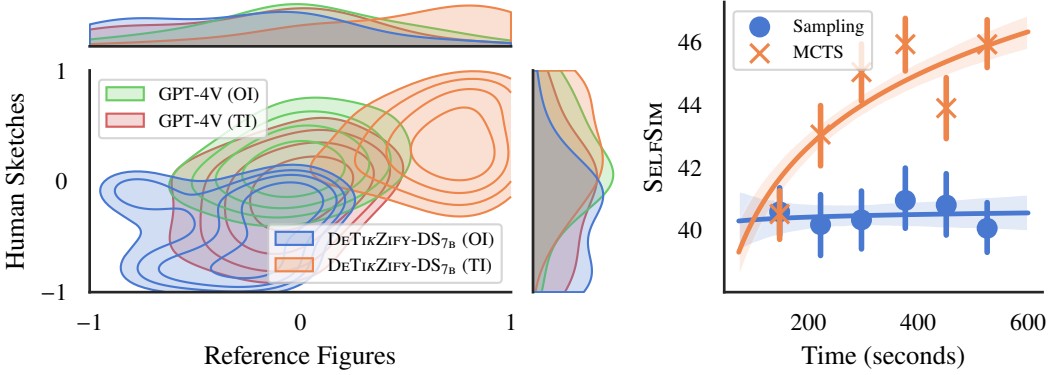

Figure 3: Bivariate distributions of BWS scores (higher is better) using kernel density estimation (left) and log-linear regression over TI reward scores for different generation strategies over time (right).

the model input type differs from the one used for evaluation. The 2.2–5.6pp increase of SᴇʟꜰSɪᴍ for all models is not surprising since it serves as the reward signal we optimize, but DʀᴇᴀᴍSɪᴍ and TED also increase by 4.9–8.7pp and 0.5–2pp, respectively, demonstrating the efficacy of our approach. While KID improves by 1–12.1 points with reference figures, it drops by 0.9–5.5 points with sketches. We believe this is because sketches often omit minor details, such as axis tick labels, which is reflected more in the output of the TI models, biasing their overall output distributions. Therefore, we consider the substantial improvement of metrics capturing instance-level similarities to be more important. For cBLEU, we observe only minor changes (less than ±0.1pp), aligning with findings that BLEU-based metrics become less effective as performance increases (Ma et al., 2019). The MST and AVG reveal that, although 1b models produce more unique outputs within the time frame compared to their larger 7b counterparts (30–36 vs. 24.1–26.2), they still fail to close the overall gap in performance, with AVG scores ranging between 0.014–0.531 compared to 0.681–0.994 for 7b models.

Overall, all DᴇTɪκZɪꜰʏ models are capable of generating compilable outputs with reasonable efficiency. Upon examination of these outputs, it becomes evident that the 7b models, particularly DᴇTɪκZɪꜰʏ-DS₇ʙ, consistently outperform both Cʟᴀᴜᴅᴇ 3 and GPT-4V, whose performance is more comparable to the 1b range. Increasing the computational budget for DᴇTɪκZɪꜰʏ further improves performance.

## 6.2 Human Evaluation

To further assess the quality of the generated figures, we perform a human evaluation on SᴋᴇᴛᴄʜFɪɢ using *Best-Worst Scaling* (BWS; Louviere et al., 2015; Kiritchenko and Mohammad, 2016, 2017). In this process, for each reference figure, we present annotators with a tuple of generated figures and ask them to identify the most and least perceptually similar figure. We then transform this data into scores ranging from -1 (poor) to 1 (excellent) by calculating the difference between the proportion of times a figure is selected as the best and the proportion of times it is chosen as the worst (Orme, 2009). To keep the workload manageable, we focus on the most promising DᴇTɪκZɪꜰʏ model (DᴇTɪκZɪꜰʏ-DS₇ʙ) and the strongest baseline (GPT-4V). Building upon the automatic evaluation, we assess these models in the OI and TI configurations, using either reference figures or human-created sketches as input. For each input type, we engage six unique expert annotators (cf. Appendix D for more details).

**Results** Figure 3 (left) shows kernel density estimates for the computed BWS scores, revealing intriguing findings that are consistent across input types. In contrast to the automatic evaluation, DᴇTɪκZɪꜰʏ-DS₇ʙ performs worse (mean score $\mu = -0.32$) than GPT-4V ($\mu = 0.09$) in OI. This could be attributed to the fact that TᴇX.SE, the sole source of SᴋᴇᴛᴄʜFɪɢ, emphasizes minimum working examples, a type on which GPT-4V particularly excels (Belouadi et al., 2024). However, when we increase the computational budget, as in DᴇTɪκZɪꜰʏ-DS₇ʙ (TI), it not only improves over OI results ($\mu = 0.39$; in line with automatic evaluation) but also surpasses GPT-4V in both configurations by a considerable margin. Interestingly, GPT-4V's performance in TI ($\mu = -0.16$) is lower than its performance in OI, indicating that GPT-4V (TI) struggles to refine its own outputs effectively and quickly deteriorates. Overall, this shows how difficult it is for models to refine their own outputs and highlights the effectiveness of our MCTS-based approach. Example outputs are provided in Table 4.

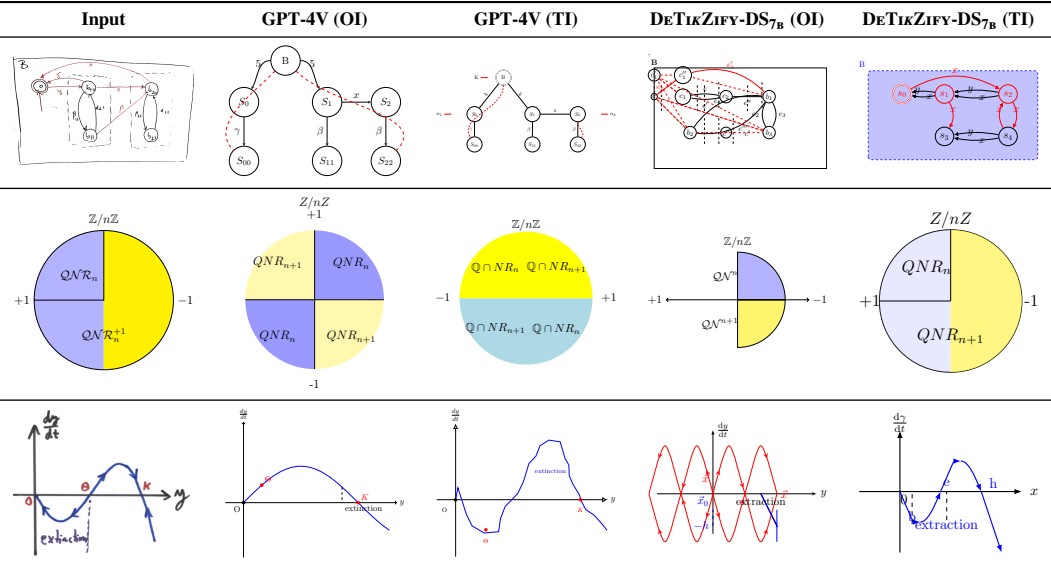

| Input | GPT-4V (OI) | GPT-4V (TI) | DETIκZIFY-DS$_{7B}$ (OI) | DETIκZIFY-DS$_{7B}$ (TI) |
|---|---|---|---|---|

Table 4: Examples of model inputs and generated outputs from our human evaluation, where annotators rated GPT-4V (OI) higher than DETIκZIFY-DS$_{7B}$ (OI) but ranked DETIκZIFY-DS$_{7B}$ (TI) as the overall best model, illustrating our findings in §6.2. See Appendix E for more examples.

## 7   Analysis

In this section, we take a closer look at our methodologies and evaluation strategies, correlating evaluation metrics with human judgments, quantifying the quality of synthetic sketches, and examining the rate of convergence of our MCTS algorithm. We also demonstrate that our models are not affected by memorization of the training data, as shown in Appendix B.

**Correlating Humans and Metrics**  To assess the reliability of our human evaluation results, we investigate the agreement between annotators. To this end, we calculate the *split-half reliability* (SHR; Kiritchenko and Mohammad, 2017) by randomly splitting our annotations into two subsets, computing BWS scores for each subset, and measuring their correlation with Spearman's $\rho$. The SHR values of 0.69 for sketches and 0.75 for images indicate a moderate to strong correlation between annotators, supporting the validity of our human evaluation results. Motivated by these findings, we explore whether metrics that also assess perceptual similarity (i.e., SELFSIM and DREAMSIM) correlate with these human judgments. We again calculate Spearman's $\rho$ and show the average correlations (David M. Corey and Burke, 1998) at the segment and system level in Table 5. For comparison, we also include the popular LPIPS and DISTS metrics (Zhang et al., 2018; Ding et al., 2020). At the segment level, SELFSIM outperforms all other metrics, which is remarkable considering it is the only untrained metric. Segment-level performance is particularly important for fine-grained reward functions, justifying our choice of SELFSIM in our MCTS algorithm. At the system level, DREAMSIM performs the best, showcasing its strength in evaluation settings.

| Metric | Segment | System |
|---|---|---|
| LPIPS | 0.224 | 0.642 |
| DISTS | 0.32 | 0.642 |
| DSIM | 0.424 | **0.954** |
| SSIM | **0.436** | 0.642 |

Table 5: Correlations of image similarity metrics with humans at the segment and system level.

**Synthetic Sketch Quality**  We also assess the quality of our synthetic sketches by measuring their congruence coefficient (Lorenzo-Seva and ten Berge, 2006) with real sketches. We embed human-created figure-sketch pairs from SKETCHFIG using SIGLIP, subtract each sketch embedding from the corresponding figure embedding to obtain *local* sketch vectors, and perform a single-component Principal Component Analysis to derive a *global* sketch vector (Zou et al., 2023). We repeat this process for synthetic sketches generated for the test split of DATIκZ$_{v2}$ and compare the global vectors using cosine similarity. Base INSTRUCT-PIX2PIX generates synthetic sketches with a congruence coefficient of 0.66, which increases to 0.7 after fine-tuning. These results demonstrate a high correlation with human-created sketches, suggesting that our generated sketches are of good quality.

**MCTS Convergence**    To gain insights into the long-term characteristics of our MCTS algorithm, we visualize the trends in achieved TI reward scores over time in Figure 3 (right) and compare them to conventional sampling-based inference. As expected, sampling does not lead to improvements over time due to the absence of a feedback loop. In contrast, MCTS consistently improves throughout the entire time frame, and even at the end of our budget of 10 minutes, it does not appear to converge, suggesting potential additional gains for larger budgets. Apart from this, MCTS is not only more effective but also faster. With an average MST of 25.17, compared to 18.7 for sampling, our MCTS algorithm generates considerably more unique TikZ programs within the same amount of time.

## 8    Conclusion

In this work, we showcase the potential of DeTikZify in generating TikZ programs for two practical use cases. First, it can convert existing figures from lower-level formats into TikZ, paving the way for semantic image editing and downstream tasks (Zhang et al., 2023a). Second, it can develop hand-drawn sketches into TikZ graphics, which could aid researchers in creating high-quality scientific illustrations. In both cases, DeTikZify substantially outperforms the commercial LLMs GPT-4V and Claude 3 despite its presumably much smaller size. We hope that our datasets (DaTikZ$_{v2}$, SketchFig, and MetaFig), our method for generating synthetic sketches, and our MCTS-based inference algorithm will pave the way towards future research on graphics program synthesis and bolster the cause of open science.

Looking ahead, we plan to extend our approach to other graphics languages, such as MetaPost, PSTricks or Asymptote (Hobby, 2014; Van Zandt, 2007; Hammerlindl et al., 2024). We also intend to explore alternatives to perceptual similarity as an MCTS reward signal, including per-pixel measures and point cloud metrics (Wang and Bovik, 2009; Wu et al., 2021). In addition, we aim to investigate reinforcement learning from reward functions, for example, using Direct Preference Optimization (Rafailov et al., 2023; Xu et al., 2024). Finally, while this work focuses on visual inputs, we plan to explore additional modalities, such as text and mixed-modality inputs, in future work.

## Limitations

In this work, we compare openly available models with proprietary systems that lack transparency in their training details and internal workings and whose performance is not stable over time. This inevitably complicates efforts to address concerns such as data leakage or cross-contamination and limits the fairness and reproducibility of our experiments. Nevertheless, under these adverse conditions, our open models and methods demonstrate favorable performance. Users should be aware, however, that our models might inherit biases, flaws, or other limitations present in the training data, potentially leading to discrepancies between expected results and generated outputs. Furthermore, given the resource-intensive nature of LLMs, many of our training and inference hyper-parameters were adopted from related work or chosen based on general intuition. Although LLMs are generally robust to hyper-parameter selection (Beyer et al., 2024), conducting a thorough hyper-parameter search might enhance their performance further. Finally, it should be noted that our models could potentially be misused by malicious actors to produce misinformation and fake science.

Another important consideration is that the public release of DaTikZ$_{v2}$ does not include some TikZ programs from our internal version due to licensing restrictions. These programs are distributed under the arXiv.org perpetual, non-exclusive license, which prohibits redistribution. Nonetheless, we provide our dataset creation scripts alongside usage instructions, enabling anyone to reproduce the full version of DaTikZ$_{v2}$ independently. The remaining TikZ programs in DaTikZ$_{v2}$ are licensed under Creative Commons attribution licenses,[5] the GNU Free Documentation License,[6] or the MIT license,[7] and their respective terms and conditions apply. Regarding artificially created examples, OpenAI's terms of use restrict the use of their services for creating competing products, limiting this subset of DaTikZ$_{v2}$ to non-commercial applications.[8]

---

[5] https://creativecommons.org/licenses
[6] https://www.gnu.org/licenses/fdl-1.3.en.html
[7] https://opensource.org/license/mit
[8] https://openai.com/policies/terms-of-use

## Acknowledgments

We would like to express our sincere gratitude to the following individuals for their contributions to our work: JiWoo Kim, Tommaso Green, Christoph Leiter, Ines Reinig, Martin Kerscher, Margret Keuper, Christopher Klamm, Daniil Larionov, Yanran Chen, Tornike Tsereteli, and Daniel Ruffinelli. Their assistance with our human evaluation campaign, proofreading, insightful discussions, and constructive comments have been invaluable. The last author is supported by the Federal Ministry of Education and Research (BMBF) via the research grant Metrics4NLG and the German Research Foundation (DFG) via the Heisenberg Grant EG 375/5–1. We would also like to acknowledge the OpenMoji project for providing the open-source icons used throughout this work and Hugging Face for their generous community GPU grant.

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

# A  Further Details on MCTS

In this section, we discuss several extensions to our MCTS algorithm that aim to improve its performance and efficiency. We also explain alternative reward functions that we experimented with but ultimately found less effective than our chosen approaches.

## A.1  MCTS Enhancements

Building on our base MCTS implementation, we introduce several enhancements, namely dynamic rescaling of visual rewards, node deduplication, and preemptive stopping of faulty rollouts.

**Dynamic Rescaling**  One challenge when using SELFSIM is that MCTS expects values to be in the range of $[-1, 1]$, while deep encoders often work with a much narrower range in practice (Hessel et al., 2021; Zhang et al., 2020). Furthermore, this range may vary depending on whether the input image is a real figure or a sketch. To address this discrepancy, we propose dynamically min-max normalizing the visual reward scores whenever they are (re)computed, ensuring that MCTS always operates on the full range. The modified reward formula is as follows:

$$V'_{i,j} = \begin{cases} \frac{V_{i,j} - \min(V_{i,:} \setminus \{-1\})}{\max(V_{i,:}) - \min(V_{i,:} \setminus \{-1\})} & \text{if } V_{i,j} \neq -1 \text{ and } \max(V_{i,:}) \neq \min(V_{i,:} \setminus \{-1\}), \\ 0 & \text{if } V_{i,j} \neq -1 \text{ and } \max(V_{i,:}) = \min(V_{i,:} \setminus \{-1\}), \\ -1 & \text{otherwise.} \end{cases} \quad (3)$$

**Node Deduplication**  During a rollout for a backtracking node, it is possible to generate code that already exists elsewhere in the tree (i.e., in siblings and their descendants). To prevent the duplication of nodes, we always merge identical node states before adding any nodes to the tree.

**Preemptive Stopping**  If the code generated in a rollout cannot be compiled due to a fatal error, we record the rollout, including the state in which the faulty line of code was first introduced. If the same (intermediate) state is sampled again during subsequent rollouts, we know that the completed output will fail to compile. In such cases, we preemptively abort the rollout and reuse the previously recorded rollout for the remainder of the simulation. To further prevent continuations from faulty code, during the expansion phase, we only add nodes to our tree whose node states do not contain any lines of code with fatal errors.

## A.2  Additional Reward Functions

Taking inspiration from popular machine translation metrics (Belouadi and Eger, 2023; Zhao et al., 2019, 2020; Song et al., 2021), which compute the Earth Mover's Distance (EMD; Rubner et al., 1998; Kusner et al., 2015) between word embeddings, we also explore with measuring perceptual image similarity as the EMD between SIGLIP's image patch embeddings. Given the distance matrix $\boldsymbol{D}$, where $D_{i,j} = \cos(x_i, y_j)$ and $\boldsymbol{x}, \boldsymbol{y}$ are the patch embedding vectors of the input and output images of simulation $j$ with lengths $|\boldsymbol{x}|$ and $|\boldsymbol{y}|$, respectively, EMD is defined as follows:

$$\text{EMD}(x, y) = \frac{\sum_{i=1}^{|\boldsymbol{x}|} \sum_{j=1}^{|\boldsymbol{y}|} F_{i,j} D_{i,j}}{\sum_{i=1}^{|\boldsymbol{x}|} \sum_{j=1}^{|\boldsymbol{y}|} F_{i,j}}, \quad \text{with} \quad \min_{\boldsymbol{F} \geq 0} \sum_{i=1}^{|\boldsymbol{x}|} \sum_{j=1}^{|\boldsymbol{y}|} F_{i,j} D_{i,j} \quad \text{s.t.} \quad \forall_{i,j} \begin{cases} \sum_{i=1}^{|\boldsymbol{x}|} F_{i,j} = \frac{1}{|\boldsymbol{y}|}, \\ \sum_{j=1}^{|\boldsymbol{y}|} F_{i,j} = \frac{1}{|\boldsymbol{x}|}. \end{cases} \quad (4)$$

We define $V_{i,j} = 2 \tanh(-\text{EMD}(x, y)) + 1 \in [-1, 1]$ if compilation produces any output. If compilation fails, we set the reward to -1. We empirically tune the hyperparameter on which layer to extract the patch embeddings using the perceptual similarity dataset of scientific figures from Belouadi et al. (2024). We find that extracting embeddings after the 24th layer yields the best results. However, when evaluated on our data, this reward function achieves a segment-level correlation of only 0.425 (cf. §7), which is lower than for SELFSIM while being computationally more expensive. Consequently, we do not employ this reward function in further experiments.

# B  Additional Experimental Results & Analyses

In Table 6, we compare LIVE (Ma et al., 2022), a state-of-the-art method for generating SVG, with our TikZ-based approach. In Figure 4, we additionally investigate the extent to which our models memorize the training data. We also perform training data ablation studies, as presented in Table 7.

Table 6: System-level scores for LIVE, an SVG-generating model, compared with TikZ-based models

| Models | Reference Figures | | | Synthetic Sketches | | |
|---|---|---|---|---|---|---|
| | DSIM↑ | SSIM↑ | KID↓ | DSIM↑ | SSIM↑ | KID↓ |
| LIVE | 57.078 | 69.253 | 324.219 | 49.455 | 64.998 | 416.016 |
| CLAUDE 3 | 64.896 | 83.372 | 17.822 | 59.102 | 73.954 | 29.541 |
| GPT-4V | 69.741 | 86.215 | 6.714 | 61.98 | 75.687 | 33.203 |
| DETiκZIFY-TL$_{1.1B}$ | 65.538 | 84.161 | 15.747 | 60.585 | 77.947 | 21.851 |
| DETiκZIFY-DS$_{1.3B}$ | 68.659 | 86.079 | 11.536 | 62.756 | 79.097 | 17.334 |
| DETiκZIFY-CL$_{7B}$ | 72.315 | 87.466 | 8.301 | 65.118 | 79.717 | 12.207 |
| DETiκZIFY-DS$_{7B}$ | 73.01 | 88.323 | 5.951 | 65.198 | 80.207 | 12.207 |

Table 6: System-level scores for LIVE, an SVG-generating model, compared with TikZ-based models from output-driven inference. Scores for TikZ-based models are copied from Table 2 for easy reference. Bold and underlined values indicate the best and second-best scores for each metric column, respectively. Cell shading reflects the relative score magnitudes across input types. Arrows indicate metric directionality.

## B.1 Comparing TikZ and SVG

Since LIVE generates SVG code instead of TikZ, we do not report cBLEU and TED scores. Additionally, because it optimizes Bézier curves rather than generating tokens, we exclude MTE, leaving only the image similarity metrics DREAMSIM, SELFSIM, and KID. Table 2 shows that LIVE underperforms all other models in our evaluation. On reference figures, it scores over 7.8pp and 14.1pp lower than the worst baseline model on DREAMSIM and SELFSIM, respectively, and its KID is more than 18 times higher. This subpar performance can be attributed to the complexity of scientific figures saved as SVGs. While we use LIVE in its default configuration, generating eight paths with four segments each, our scientific figures consist of over 110 paths on average with an arbitrary number of segments, not counting deduplicated paths, which LIVE cannot detect. Although we could theoretically configure LIVE to generate more paths, this would linearly increase inference time, quickly becoming intractable. LIVE already requires over 18

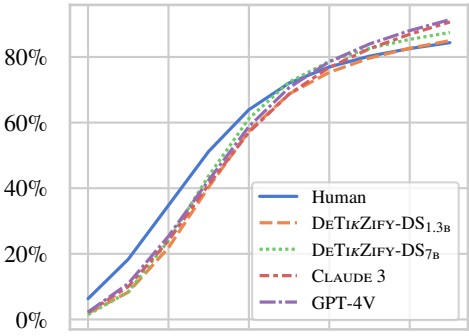

Figure 4: Proportion of generated code $n$-grams with $n \in [1, 10]$ that are novel (i.e., not present in the training data). Results for human-created code are included as a reference point for comparison.

hours to complete the test set for one input type, whereas DETiκZIFY-CS$_{7B}$ (OI), for example, takes less than 5 hours. Furthermore, since LIVE attempts to vectorize the input directly without semantic interpretation, it performs even worse on synthetic sketches. We conclude that SVG, and, by extension, models that generate SVG, are not well-suited for our problem domain and objectives.

## B.2 Memorization

Memorization of training data is a common concern in language models (McCoy et al., 2023; Carlini et al., 2023; Raunak and Menezes, 2022; Meehan et al., 2020). To assess the extent of this issue in our models, we calculate the *n-gram novelty* (McCoy et al., 2023). Specifically, we determine the proportion of $n$-grams, with $n \in [1, 10]$, in the model-generated TikZ programs that are *not* present in the training data. We perform this analysis on the test split of DATiκZ$_{v2}$ for our baselines and DEEPSEEK-based DETiκZIFY models conditioned on reference figures, as well as human-generated code, as shown in Figure 4. All models initially exhibit similar novelty and are slightly less novel than humans for $n < 7$. However, starting from $n = 7$, all models except DETiκZIFY-DS$_{1.3B}$ surpass human novelty, with more than 80% of all model-generated $n$-grams being novel for $n >= 8$. This phenomenon of models becoming more novel than humans is commonly observed and is considered an indicator that language models are not significantly affected by memorization (McCoy et al.,

| | Reference Figures | | | | | | Synthetic Sketches | | | | | |
|---|---|---|---|---|---|---|---|---|---|---|---|---|
| **Models** | **MTE$_\uparrow$** | **cBLEU$_\uparrow$** | **TED$_\downarrow$** | **DSim$_\uparrow$** | **SSim$_\uparrow$** | **KID$_\downarrow$** | **MTE$_\uparrow$** | **cBLEU$_\uparrow$** | **TED$_\downarrow$** | **DSim$_\uparrow$** | **SSim$_\uparrow$** | **KID$_\downarrow$** |
| Full Training | 83.771 | 1.336 | 57.661 | 68.659 | 86.079 | 11.536 | 87.446 | 0.541 | 60.112 | 62.756 | 79.097 | 17.334 |
| −Synthetic Sketches | −1.957 | +0.327 | −0.822 | +2.433 | +1.318 | +3.296 | −13.358 | +0.171 | +1.369 | −3.993 | −3.332 | +34.18 |
| −MᴇᴛᴀFɪɢ | −1.846 | −0.096 | −0.356 | +0.398 | −0.046 | +0.115 | −0.132 | +0.053 | −0.378 | +0.084 | −0.181 | +2.773 |

Table 7: Ablation study results for DT-TL$_{1.1\text{B}}$ (OI), showing the relative impact on test set performance when either sketch-based training or connector pretraining is omitted, compared to full training. Improvements are highlighted in green, and declines in red, with reference scores taken from Table 2.

2023; Belouadi et al., 2024). Interestingly, for larger $n$-grams, DᴇTɪᴋZɪғʏ-DS$_{7\text{B}}$ demonstrates higher novelty than its smaller counterpart, suggesting that despite its larger capacity, it does not overfit and generalizes well. The most novel models are GPT-4V and Cʟᴀᴜᴅᴇ 3, possibly because they were not trained on DᴀTɪᴋZ$_{\text{v2}}$ and might have been trained on data that has been prepared differently.

## B.3 Training Data Ablation Studies

To better understand the impact of training with synthetic sketches and pretraining using MᴇᴛᴀFɪɢ on test set performance, we conducted ablation studies with DᴇTɪᴋZɪғʏ-DS$_{1.3\text{B}}$ in the OI configuration as a representative model, following the experimental setup detailed in §6.1. In particular, Table 7 compares full training with variations where synthetic sketches are excluded and the step of pretraining the connector is omitted. The results from excluding synthetic sketches align with expectations: although this approach slightly improves performance on reference figures on average, it substantially reduces performance on sketches. Therefore, for models expected to perform well on both figures and sketches, we recommend our original training methodology. Conversely, for models focused solely on figures, training exclusively on figures may be advantageous. The findings related to skipping connector pretraining are less definitive as the score differences are minimal, reflecting the lack of consensus in related literature about the benefits of connector pretraining for downstream performance (Liu et al., 2023b,a; Karamcheti et al., 2024). However, on average, we observe a positive impact, especially on MTE and KID, where consistent improvements are noted for both reference figures and synthetic sketches as input. Thus, we advocate incorporating a dedicated pretraining step in the training protocol. In future work, we also plan to investigate the impact of pretraining dataset size and quality.

## C Additional Training & Inference Details

In this section, we provide supplementary information on the training and inference procedures for all our models. For training and inference of our local DᴇTɪᴋZɪғʏ models, we utilize a compute node equipped with four Nvidia A40 GPUs and 448 gigabytes of RAM. We access Cʟᴀᴜᴅᴇ 3 and GPT-4V through their respective official API endpoints.

### C.1 DᴇTɪᴋZɪғʏ

Complementing the information provided in §4, our 1b models require approximately two days of fine-tuning on our hardware. For the 7b models, we employ optimizer state and gradient partitioning (Rajbhandari et al., 2020) to accommodate them within the available resources, resulting in an extended training time of 21 days. Generating sketches for the training runs takes an additional 1.5 days, but since we cache our sketches, these costs are incurred only once. Output-driven inference takes 4–8 hours, depending on the model and input type, and time-budgeted inference extends the runtime by a further 1.5 days.

### C.2 Iɴsᴛʀᴜᴄᴛ-Pɪx2Pɪx

As SᴋᴇᴛᴄʜFɪɢ with only 549 examples may be considered too small for fine-tuning Iɴsᴛʀᴜᴄᴛ-Pɪx2Pɪx, we augment our training data with 4000 additional sketches of natural images (Sangkloy et al., 2016; Li et al., 2019) and 2000 synthetic sketches of scientific figures generated with base Iɴsᴛʀᴜᴄᴛ-Pɪx2Pɪx. We then oversample SᴋᴇᴛᴄʜFɪɢ at a 5:1 ratio and, following Paul (2023), train for 15k steps with a

batch size of 8 and a learning rate of 5e−5. We select "turn it into a doodle" as our initial prompt, which also appears in INSTRUCT-PIX2PIX's pretraining dataset and demonstrates the most promising zero-shot performance.

### C.3  CLAUDE 3 & GPT-4V

Building upon the experiments described in §6, we derive all our Self-Refine prompts from the official examples provided by Madaan et al. (2023) for generating Ti*k*Z programs, with only minor modifications. In particular, we employ the following prompt template in the initial step of both Self-Refine and Visual Self-Refine, substituting "sketch" or "picture" as appropriate:

```
1  This is a [ sketch | picture ] of a scientific figure.  Generate
2  LaTeX code that draws this scientific figure using TikZ. Ensure
3  that the LaTeX code is self-contained and does not require any
4  packages except TikZ-related imports.  Don't forget to include
5  \usepackage{tikz}!  I understand that this is a challenging task,
6  so do your best.  Return your result in a ```latex code block.
```

We then extract the first LaTeX code block from the generated text. In the rare cases where GPT-4V incorrectly classifies input images as unsafe, we add a small amount of Gaussian noise to the image pixels to bypass the issue. If compilation fails due to a fatal error (which occurs in only 1.5% of all cases) without producing an output artifact, we repeatedly use the following prompt template until all issues are resolved, replacing `` with the generated code and `<error>` with the corresponding error message:

```
1  Given the error message:
2  <error>
3  And the problematic code:
4  ```latex
5  
6  ```
7  First, identify the issue based on the error message.  Then,
8  determine the cause of the error in the code.  Finally, propose
9  and implement a solution.  Return the fixed code in a ```latex code
10 block.
```

For Visual Self-Refine, we additionally use the following prompt template to visually refine the output. Since we provide two input images (the initial figure or sketch and the current output), we label one as "Input" and the other as "Reference". CLAUDE 3's API has a built-in mechanism for labeling images, while for GPT-4V, we embed the labels directly into the images:

```
1  ```latex
2  
3  ```
4  This is the TikZ/LaTeX code for the scientific figure shown in the
5  picture labeled "Input".  Can you improve it to better resemble
6  the provided reference [ sketch | picture ]?  First, analyze the
7  "Input" picture to understand its components and layout.  Then,
8  consider how the scientific figure can be enhanced to more closely
9  match the reference [ sketch | picture ].  Finally, rewrite the
10 TikZ code to implement these improvements, making the image more
11 similar to the reference.  Ensure that the LaTeX code is self-
12 contained and does not require any packages except TikZ-related
13 imports. Don't forget to include \usepackage{tikz}! Return your
14 result in a ```latex code block.
```

Following the findings of Madaan et al. (2023), we visually refine for a maximum of four iterations, as they observe diminishing returns beyond that point, and it helps reduce inference costs. Although this means that in most cases, we terminate before the 10-minute timeout is reached (cf. §6.1), we believe this is a sensible decision, as we observe that GPT-4V is unable to visually refine its

outputs successfully in any case. We hypothesize that this limitation is due to general-purpose chat models requiring too much explicit context for this task. These models receive the entire previously generated code as input, along with two input images and a complex textual prompt, which may be too challenging for them to process effectively. Preliminary experiments with more elaborate prompts did not seem to mitigate the subpar performance, likely due to this reason.

## D Annotator Demographics

Our annotator team consists of eleven experts with extensive research experience in science and technology. The team comprises one male faculty member, two female PhD students, seven male PhD students, and one male research assistant from another institution. We chose to work exclusively with expert annotators based on the findings of Belouadi et al. (2024), which demonstrated that crowd annotators often lack the necessary research background to produce reliable annotations.

## E Examples

To provide a better understanding of our work, we present a variety of examples in this section. Table 8 displays exemplary figures and real sketches from SKETCHFIG, while Table 9 shows figures and synthetic sketches from DATiκZv2. Additionally, Tables 10 & 11 present sample outputs generated by our systems during our human and automatic evaluations. Figure 5 provides a closer look at generated code.

When comparing the real sketches in Table 8 to their corresponding reference figures, it becomes evident that the sketches often contain less detail. For instance, sketches may lack colors or grids and feature less precise lines. Moreover, the handwritten nature of the sketches can sometimes make the text within them harder to read. These characteristics are also present in the synthetic sketches shown in Table 9. However, the problem of illegible text is more pronounced in these sketches, as generating readable text remains a common challenge for image generation models (Borji, 2023). While the text may still retain its meaning in a hidden way (Daras and Dimakis, 2022), this could lead to hallucinated text in the generated TiκZ programs. Nonetheless, we believe that this aspect can still be advantageous for end users, as it enables them to quickly add scribbles to indicate the desired text placement. By doing so, DETiκZIFY can generate code for the overall structure and layout, allowing users to easily modify and replace the text afterward.

The randomly selected generated figures from our human and automatic evaluations (cf. §6.2 & §6.1) shown in Tables 10 & 11 corroborate our quantitative findings. DETiκZIFY-DS$_{7B}$ (TI) demonstrates the best overall performance and shows the least amount of fidelity errors, confirming the effectiveness of our SELFSIM-based MCTS refinement algorithm. However, we still observe some inconsistencies, such as in layout and axes labeling, although to a lesser extent compared to DETiκZIFY-DS$_{7B}$ (OI) and GPT-4V. We attribute the prevalence of this problem partly to our focus on perceptual similarity rather than, e.g., pixel-level similarity, which allows the models greater flexibility in interpreting the general semantics of the input figures and sketches. While optimizing pixel-level similarity could be an alternative approach, we argue that perceptual similarity can serve as a more meaningful measure, especially when considering sketches. We believe that real users who provide rough sketches of unfinished ideas will find the generated outputs that interpret and refine their concepts to be inspirational. However, we acknowledge the potential benefits of exploring more rigorous similarity measures and plan to investigate this in future research. Interestingly, GPT-4V occasionally generates outputs that may not be appropriate in a scientific context, such as mistakenly embedding a smiley face in the fourth example in Table 10. Instead of resolving such issues, GPT-4V (TI) further emphasizes these details, distancing the output from the actual reference.

Figure 5 provides a side-by-side comparison of the generated TiκZ programs corresponding to the first row in Table 10. DETiκZIFY-DS$_{7B}$ demonstrates its ability to utilize advanced abstractions and control flow statements, generating code that is free of compile-time errors in both OI and TI configurations. On the other hand, GPT-4V (OI) incorrectly uses an undefined arrow tip kind `stealth'` in lines 9 and 10, resulting in recoverable compile-time errors. GPT-4V (TI) contains the same error in line 8 and introduces additional errors in lines 16 and 26, where the * symbol would have to be removed from the loop lists for successful expression evaluation.

| Reference Figures | Real Sketches |
|:---:|:---:|

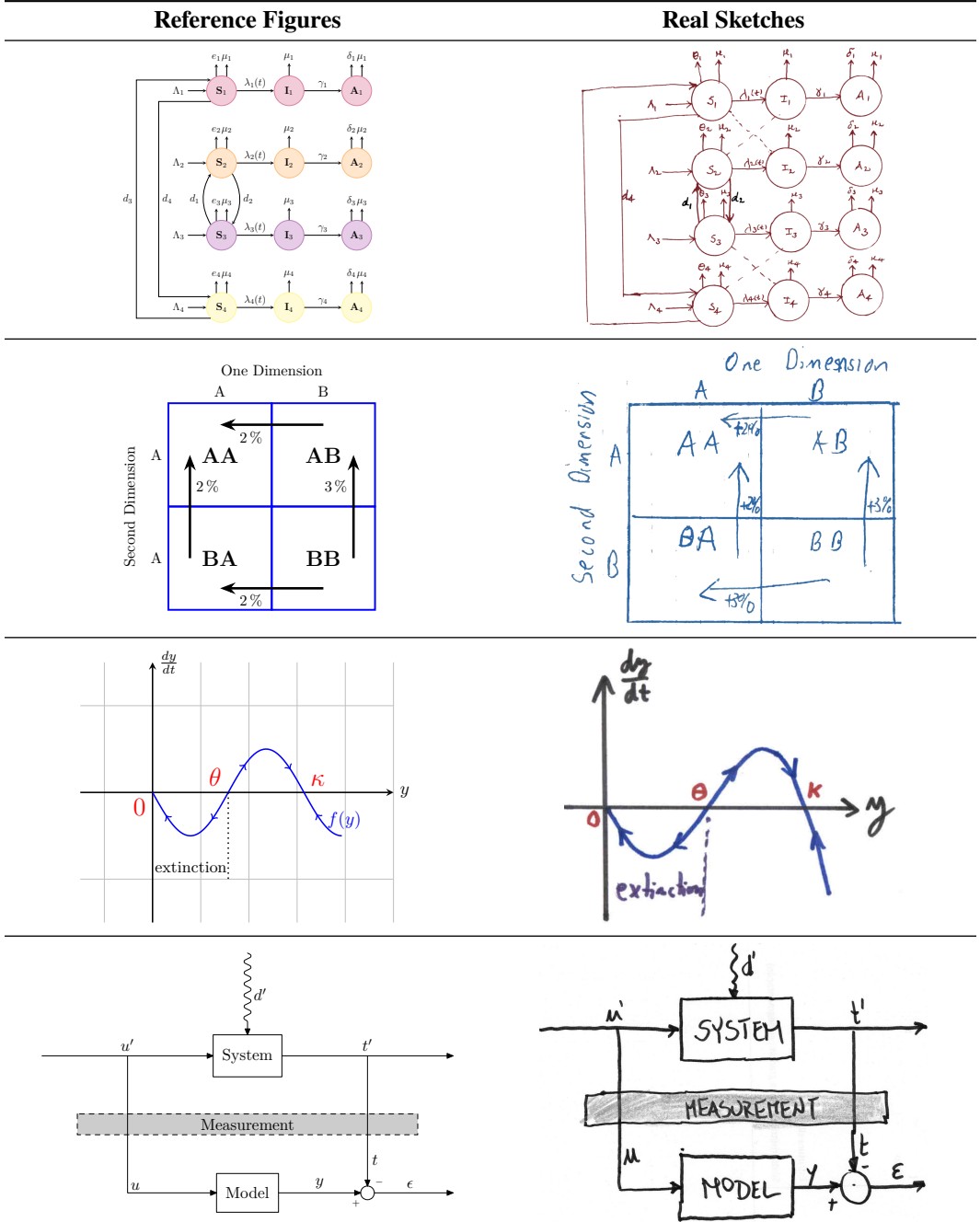

Table 8: Representative examples of reference figures paired with real sketches from the SKETCHFIG dataset.

| **Reference Figures** | **Synthetic Sketches** |
|:---:|:---:|

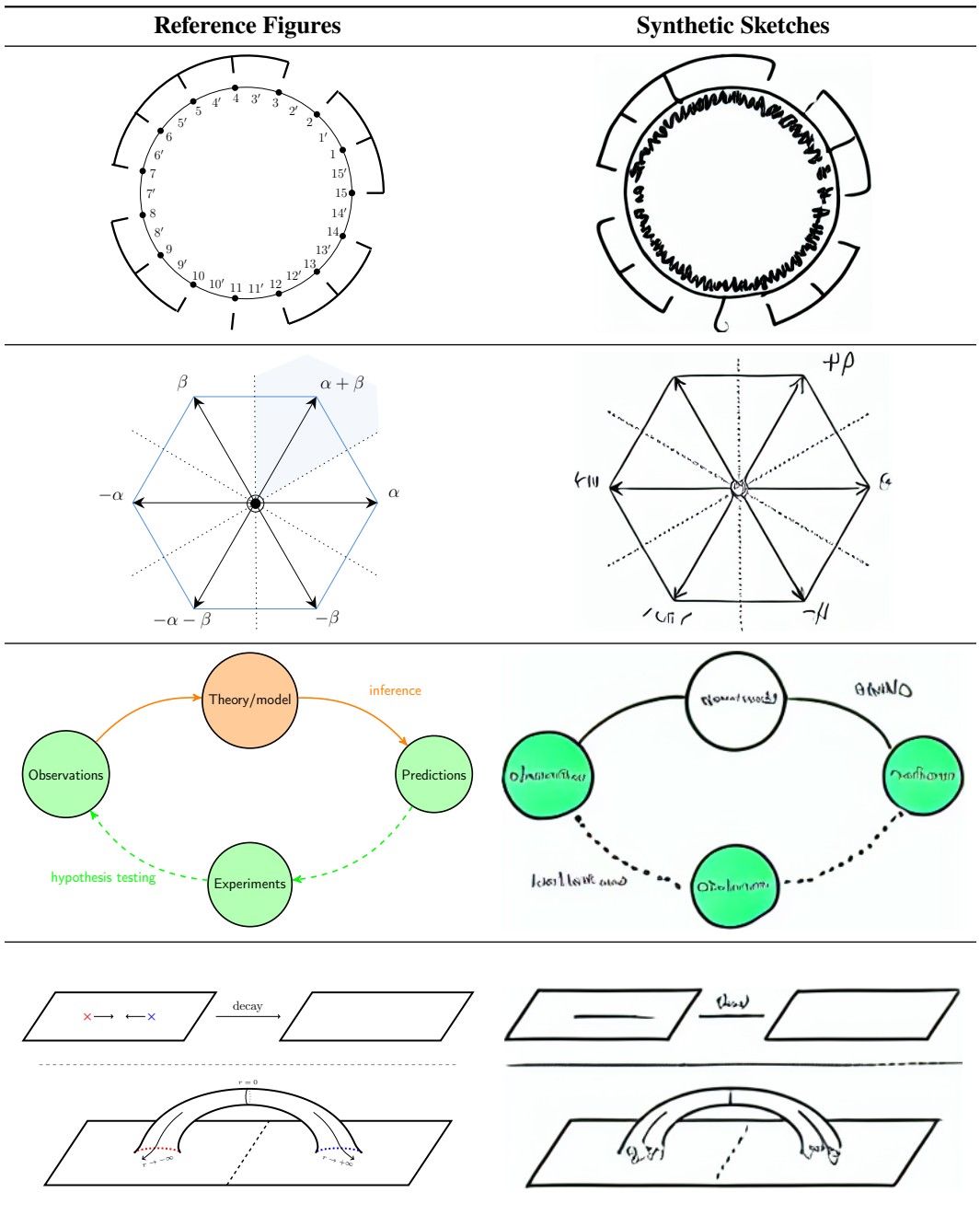

Table 9: Illustrative examples of reference figures and corresponding synthetic sketches from the subset of the DATıκZₓ₂ dataset that is licensed for redistribution.

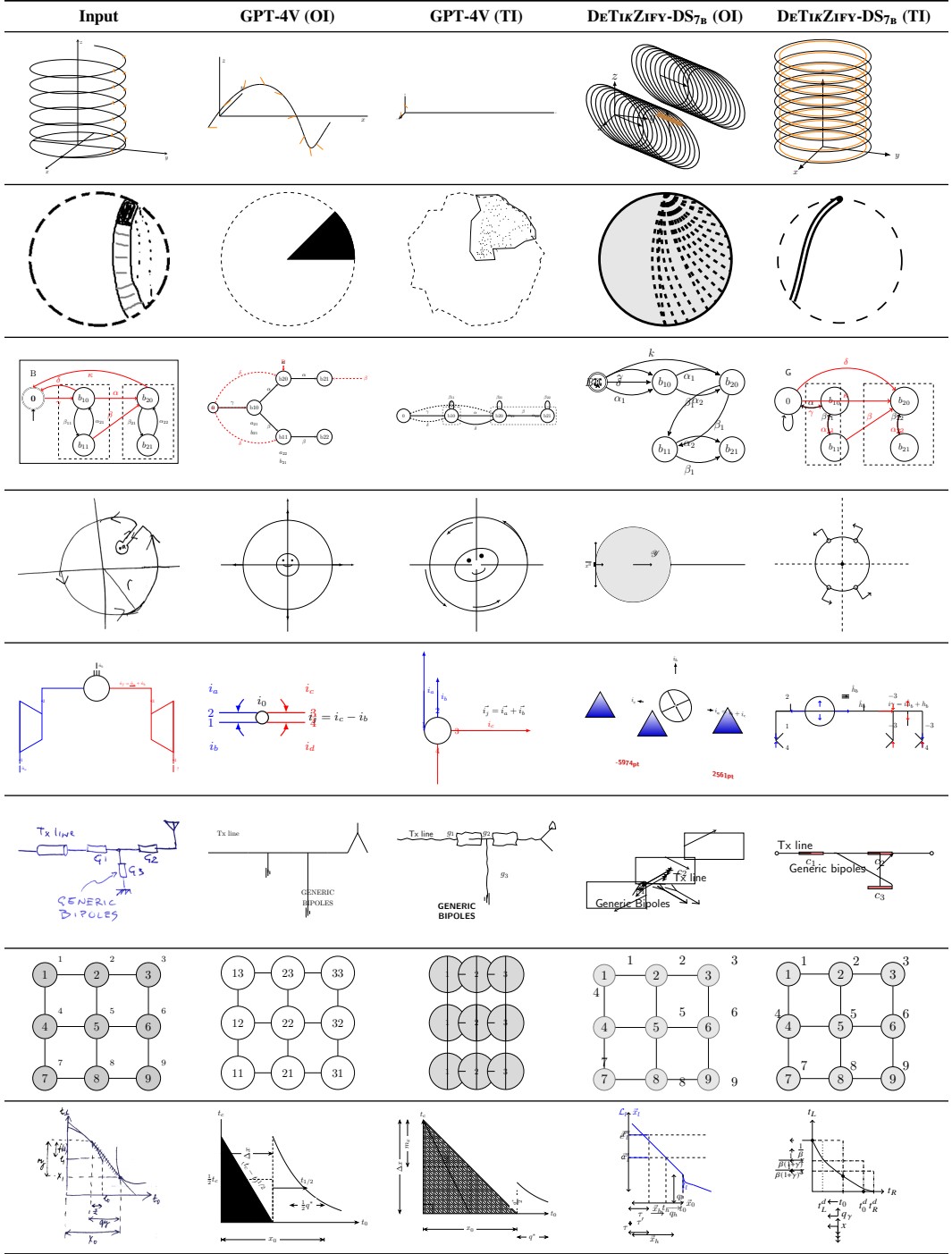

Table 10: Alternating rows of randomly selected reference figures and real sketches (first column) alongside corresponding scientific figures generated by GPT-4V and DeTiκZify-DS$_{7B}$ in output-driven (OI) and time-budgeted (TI) configurations (columns 2–4), taken from our human evaluation campaign (cf. §6.2).

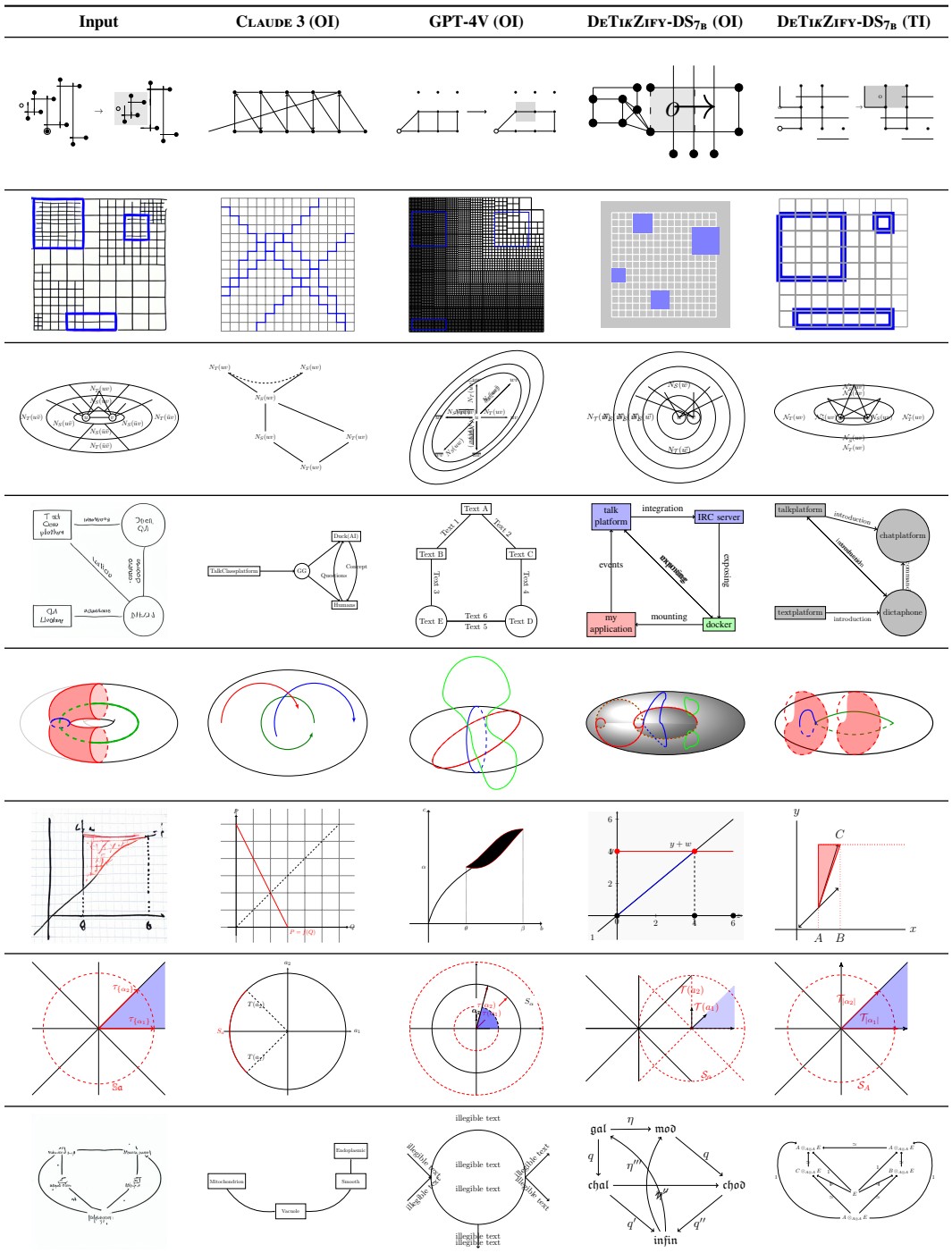

| Input | Claude 3 (OI) | GPT-4V (OI) | DeTikZify-DS$_{7B}$ (OI) | DeTikZify-DS$_{7B}$ (TI) |
|---|---|---|---|---|

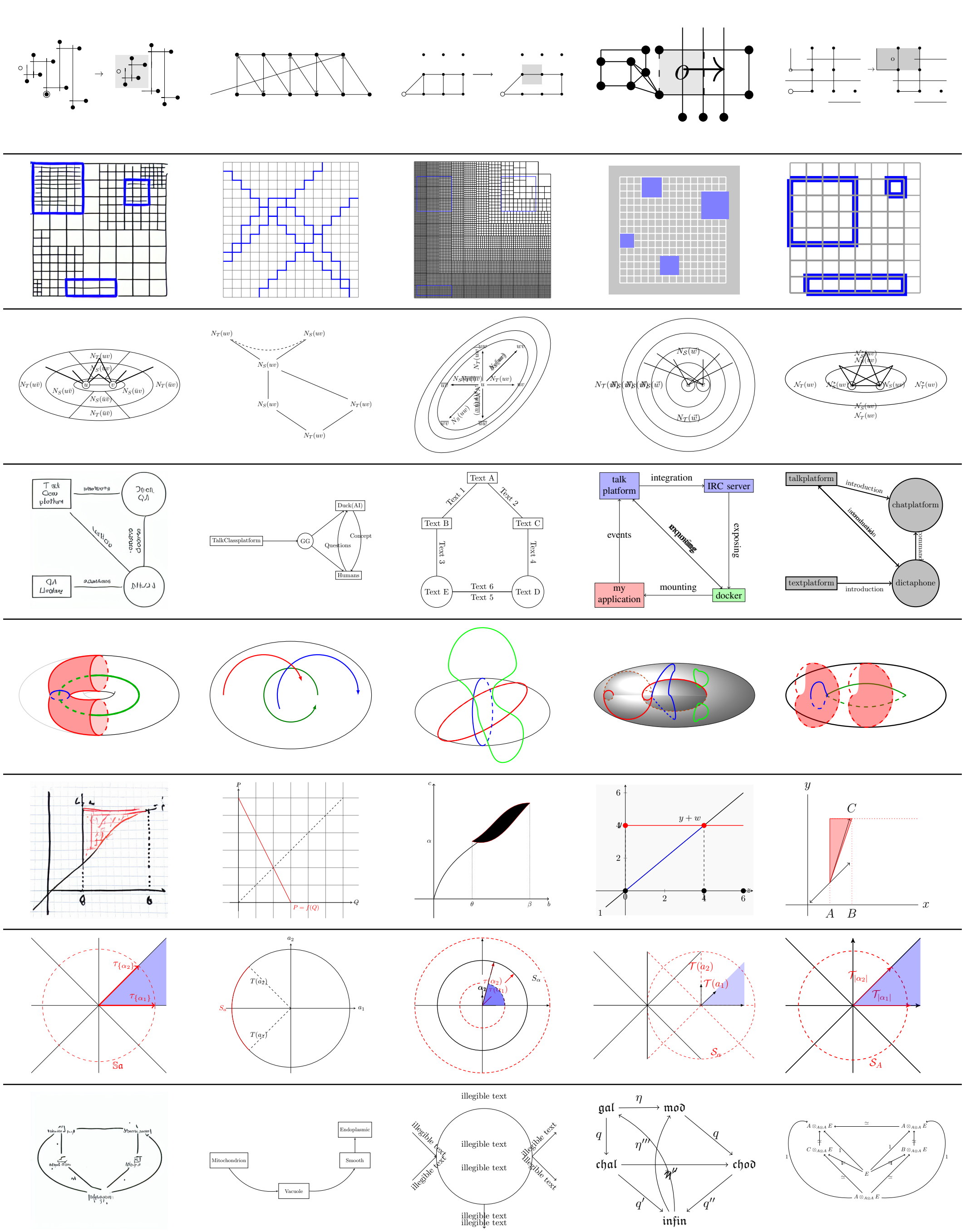

Table 11: Alternating rows of randomly selected reference figures and synthetic sketches (first column) alongside corresponding scientific figures generated by Claude 3 (OI), GPT-4V (OI), and DeTikZify-DS$_{7B}$ (OI & TI) in columns 2–4, taken from our automatic evaluation (cf. §6.1).

```latex
\documentclass[border=3pt,tikz]{standalone}
\usepackage{tikz-3dplot}
\begin{document}

\tdplotsetmaincoords{70}{110}
\tdplotsetrotatedcoords{0}{20}{70} %<- rotated coords

\begin{tikzpicture}[tdplot_main_coords,line join=round]
\pgfkeys{/pgf/number format/.cd,precision=2,fixed}

\def\n{12}
\def\h{2}

\begin{scope}[tdplot_rotated_coords,canvas is xz plane at y=0]
\draw[-latex] (\h,0) -- (\h,1);
\draw[-latex] (0,0) -- (1,0);
\foreach \i in {0,...,\n}
{
   \draw[thick] (\i*\h/\n,0) circle[radius=1];
}
\end{scope}

\foreach \i in {1,...,\n}
{
   \draw[-latex,thick,brown] (\h*\i/\n,\h,\i/\n) -- (\h*\i/\n,\h,\i/\n+1/\n);
}

\begin{scope}[tdplot_rotated_coords,canvas is xz plane at y=\h]
\draw[-latex] (0,0) -- (1,0);
\foreach \i in {0,...,\n}
{
   \draw[thick] (\i*\h/\n,0) circle[radius=1];
}
\end{scope}

\draw[-latex] (0,0,0) -- (1,0,0) node[pos=1.1]{$x$};
\draw[-latex] (0,0,0) -- (0,1,0) node[pos=1.1]{$y$};
\draw[-latex] (0,0,0) -- (0,0,1) node[pos=1.1]{$z$};
\end{tikzpicture}

\end{document}
```

DeTikZify-DS$_{7B}$ (OI)

```latex
\documentclass[tikz,border=3.14mm]{standalone}
\usepackage{tikz-3dplot}
\usetikzlibrary{arrows.meta,bending}

\begin{document}

\tdplotsetmaincoords{70}{110}
\begin{tikzpicture}[tdplot_main_coords,thick]
 \draw[-{Latex[bend]}] (0,0,0) -- (3,0,0) node[pos=1.1]{$x$};
 \draw[-{Latex[bend]}] (0,0,0) -- (0,3,0) node[pos=1.1]{$y$};
 \draw[-{Latex[bend]}] (0,0,0) -- (0,0,3) node[pos=1.1]{$z$};
 \foreach \X [remember=\X as \LastX (initially 0)] in {1,...,9}
 {\begin{scope}[canvas is xy plane at z=\LastX/2]
  \draw circle[radius=2];
  \ifnum\X=1
   \draw[-{Latex[bend]},orange,shorten >=-1pt] circle[radius=1.8];
  \else
   \draw[orange,shorten >=-1pt] circle[radius=1.8];
  \fi
 \end{scope}}
\end{tikzpicture}
\end{document}
```

DeTikZify-DS$_{7B}$ (TI)

```latex
\documentclass[tikz,border=10pt]{standalone}
\usepackage{tikz}
\usepackage{amsmath}
\usetikzlibrary{arrows.meta,decorations.markings}

\begin{document}
\begin{tikzpicture}[scale=1.5]
   % Styles for arrows
   \tikzset{axis/.style={thick,->,>=stealth'}}
   \tikzset{vector/.style={thick,->,>=stealth',orange}}

   % Draw helix
   \foreach \t in {0,10,...,360} {
      \draw[domain=\t:\t+10,smooth,variable=\x,black]
         plot ({\x/360*4},{sin(\x)},{cos(\x)});
   }

   % Draw axis
   \draw[axis] (0,0,0) -- (5,0,0) node[anchor=north east]{$x$};
   \draw[axis] (0,0,0) -- (0,2,0) node[anchor=north west]{$z$};
   \draw[axis] (0,0,0) -- (0,0,-2) node[anchor=south]{$y$};

   % Draw vectors
   \foreach \t in {0,40,...,320} {
      \draw[vector] ({\t/360*4},{sin(\t)},{cos(\t)}) -- ++({cos(\t)*0.3},{sin(\t)*0.3},0);
   }

\end{tikzpicture}
\end{document}
```

GPT-4V (OI)

```latex
\documentclass[tikz,border=10pt]{standalone}
\usepackage{tikz}
\usetikzlibrary{3d,calc,arrows.meta}

\begin{document}
\begin{tikzpicture}[scale=1.5]
   % Styles for axes and vectors
   \tikzset{axis/.style={thick,->,>=stealth'}}
   \tikzset{vector/.style={thin,->,>=latex,orange}}
   \tikzset{helix/.style={line width=1mm, black, -{Triangle[width=3mm,length=3mm]}}}

   % Define the number of turns for the helix
   \def\turns{3}

   % Draw helix with arrowheads
   \foreach \t in {5,10,...,\turns*360} {
      \draw[helix] ({(\t-5)/360*4},{sin(\t-5)},{cos(\t-5)})
         -- ({\t/360*4},{sin(\t)},{cos(\t)});
   }

   % Draw axes
   \draw[axis] (0,0,0) -- (4*\turns+1.5,0,0) node[anchor=west]{$x$};
   \draw[axis] (0,0,0) -- (0,1.5,0) node[anchor=south]{$y$};
   \draw[axis] (0,0,0) -- (0,0,1.5) node[anchor=east]{$z$};

   % Draw normal vectors
   \foreach \t in {0,45,...,\turns*360} {
      % Calculate the normal vector components
      \pgfmathsetmacro{\dx}{-sin(\t)*0.5}
      \pgfmathsetmacro{\dy}{cos(\t)*0.5}
      \pgfmathsetmacro{\dz}{0}

      % Draw the normal vector
      \draw[vector] ({\t/360*4},{sin(\t)},{cos(\t)}) -- ++(\dx,\dy,\dz);
   }
\end{tikzpicture}
\end{document}
```

GPT-4V (TI)

Figure 5: Ti*k*Z programs generated by DeTikZify-DS$_{7B}$ (top) and GPT-4V (bottom) corresponding to the figures in the first row of Table 10. Lines with compile-time errors are highlighted in yellow.

