# OpenReview forum: "DeTikZify: Synthesizing Graphics Programs for Scientific Figures and Sketches with TikZ"
_NeurIPS.cc/2024/Conference — NeurIPS 2024 spotlight_

### Official Review · Reviewer_tgj9 · 2024-07-09

**Soundness:** 4
**Presentation:** 4
**Contribution:** 4
**Rating:** 8
**Confidence:** 3

**Summary:**

The paper proposes a technique for generating TikZ graphics based on handwritten sketches. It open-sources the datasets used for this task, including a large TikZ dataset, a small sketch-to-TikZ dataset, and a large dataset of scientific figures with accompanying texts.

The model uses a vision-language model to take as input an image of the sketch, and potentially the output of the LaTeX compiler from the previous step, to generate the code that is attempted for compilation in the successive step. To further improve the model, Monte Carlo Tree Search is employed for more efficient iteration in finding a compilable and high-quality output.

The evaluation is done via several automated metrics such as code and image similarity, as well as wall time for generation, accompanied by the human evaluation, which positively correlated with the used automated metrics.

**Strengths:**

The paper is clearly written, the experimentation setup, ablation and analysis is of the high quality, the task of sketch-to-tikz is somewhat novel, and it has high significance, thanks to the datasets being shared, and the bits about the generation of synthetic dataset using Instruct-Pix2Pix and using MCTS to improve the quality are valuable for many other applications, and having the openly shared code for this is also a great contribution.

**Weaknesses:**

I think it would strengthen the paper to provide more examples of the model inference (similar to page 29 of the Appendix) to better gauge the performance of the models and to see examples without MCTS to understand how much does it contribute to the downstream quality (Fig. 3, right, paints some picture, but the SelfSim metric is hard to interpret). More generally, ablation study likely can also be strengthened by highlighting the amount of contribution brought in my MCTS, synthetic sketches, SciCap++, etc.

**Questions:**

No particular questions, see suggestions in "weaknesses".

**Limitations:**

The authors have adequately addressed the limitations in the checklist.

---

> ### Author Rebuttal · Authors · 2024-08-06
>
> We thank the reviewer and are delighted to see that they generally like our work.
>
> > I think it would strengthen the paper to provide more examples of the model inference [...]. More generally, ablation study likely can also be strengthened by highlighting the amount of contribution brought in my MCTS, synthetic sketches, SciCap++, etc.
>
> We agree with the reviewer that providing more examples would be helpful, an opinion also shared by reviewer uo3w. We therefore provide more randomly selected examples which also exemplify the contributions brought by MCTS in the pdf of the general response (we will also add this to the revised version of our paper). However, we want to remark that since MCTS only has an effect on image similarity when the SelfSim reward is used, comparing examples with and without MCTS is equivalent to comparing examples from output-driven inference with time-budgeted inference. This also means that we can refer the reviewer to Sections 6.1 (image similarity metrics) and 6.2 for a quantitative comparison. We will clear this up in our revised version.
>
> To assess the impact of synthetic sketches on training, we conducted an additional ablation study using DeTikZify-DS-1.3b, comparing fine-tuning with a 50% random replacement of figures with sketches (w/ sketches) against no sketch replacement (w/o sketches):
>
> |              | Figures  |          |          |          |          |          | Sketches |          |          |          |          |          |
> | ------------ | -------- | -------- | -------- | -------- | -------- | -------- | -------- | -------- | -------- | -------- | -------- | -------- |
> |**Training**  |**MTE**   |**cBLEU** |**TED**   |**DSim**  |**SSim**  |**KID**   |**MTE**   |**cBLEU** |**TED**   |**DSim**  |**SSim**  |**KID**   |
> | w/ sketches  |**83.771**|  1.336   |  57.661  |  68.659  |  86.079  |**11.536**|**87.446**|  0.541   |**60.112**|**62.756**|**79.097**|**17.334**|
> | w/o sketches |  81.814  |**1.663** |**56.839**|**71.092**|**87.397**|  14.832  |  74.088  |**0.712** |  61.481  |  58.763  |  75.765  |  51.514  |
>
> The results align with expectations: omitting sketches slightly enhances performance on reference figures but significantly diminishes performance on sketches. For a model that should perform adequately on both tasks, our original training recipe is recommended, whereas for figure-only input, exclusive training on figures may be preferable. These findings will be
> integrated into our revised paper.
>
> Regarding the impact of pre-training on SciCap++, we direct the reviewer to existing literature on MLLM development, which shows that pre-training vision-language adapters generally improves downstream performance \[1,2,3]. Nonetheless, we agree on the importance of independent verification and will train a model without this pre-training step and include the results in the revised version of our paper.
>
> \[1]: Tong, Shengbang, et al. "Cambrian-1: A fully open, vision-centric exploration of multimodal llms." arXiv preprint arXiv:2406.16860 (2024).
>
> \[2]: Liu, Haotian, et al. "Visual instruction tuning." Advances in neural information processing systems 36 (2024).
>
> \[3]: Liu, Haotian, et al. "Improved baselines with visual instruction tuning." Proceedings of the IEEE/CVF Conference on Computer Vision and Pattern Recognition. 2024.

---

### Official Review · Reviewer_uo3w · 2024-07-12

**Soundness:** 4
**Presentation:** 4
**Contribution:** 4
**Rating:** 8
**Confidence:** 4

**Summary:**

This paper introduces DeTikZify, a multimodal model that automatically generates TikZ code for scientific figures and sketches. The authors create and leverage three novel datasets: DaTikZv2 (a large dataset of TikZ graphics), SketchFig (paired human-drawn sketches and scientific figures), and SciCap++ (a meta-dataset of scientific figures and associated text). They demonstrate that DeTikZify outperforms GPT-4V and Claude 3. They also show that a novel MCTS-based iterative refinement algorithm brings gains.

**Strengths:**

Originality: This paper introduces DeTikZify, a novel approach to automatically generate TikZ code from both existing figures and hand-drawn sketches. Prior work mostly explored image vectorization and image-to-LaTeX conversion, but this paper tackles the underexplored problem of synthesizing semantically TikZ programs from visual input. This is a valuable contribution to the field. Although the individual components are not novel, their combination is (SigLIP, CodeLLaMA/TinyLLaMA/DeepSeek and MCTS-based refinement algorithm).
The authors introduce three new datasets which are increasingly valuable contributions in the field. I also find several ideas in the paper, e.g. to randomly replace figures with 50% synthetic sketches very interesting, original and practical.
Quality: I find the scientific standards of this paper to be very high. Firstly, it provides a thorough evaluation of DeTikZify on various metrics, including code similarity, image similarity, and efficiency. The human evaluation further strengthens the findings. The ablations are insightful and further support the design choices.
Clarity: The paper is very well-structured, clearly written and easy to follow. The figures aid understanding of the method and the results.
Significance: This paper tackles a very practical problem that can have a large impact on the workflows of figure creation, editing, etc. DeTikZify is a valuable step towards improving this process. Making the code, models and datasets publicly available enables further research and development and aids everyone working in this area.

**Weaknesses:**

1. The main issue that I have with this work is the potential data leakage. It is not clear how much of the training / evaluation data is in the training set of models. How can we be sure about this?
For the public models, it is said “we only include items created after the cut-off date of CodeLLaMA and exclude repositories that may have been used in training DeepSeek.” but how to ensure this for the private models? I feel that using the “𝑛-gram matching algorithm” does not really tell us a lot about this. It would be great if the authors can provide some further thoughts / explanations on this.
2. From the text I find it hard to understand the intuition behind and the exact computation of “Average Similarity” (L249). Would it be possible to improve clarify this in the main text?

**Questions:**

1. The paper focuses on scientific figures, which often exhibit structured layouts and simple shapes. However, TikZ's capabilities extend to much more complex diagrams and illustrations. How well would DeTikZify generalize to more diverse TikZ use cases beyond scientific figures? Are there inherent limitations in the approach that hinder its application to more free-form diagrams?
2. While the paper argues for the importance of segment-level performance, it's unclear how this translates to actual user experience. Can the authors elaborate on how segment-level improvements manifest in visually perceptible differences in the generated figures?
3. The paper mentions that SelfSim outperforms other metrics in segment-level correlation. However, the difference in Spearman's ρ appears relatively small (Table 4). Could the authors comment on the practical significance of this difference?
4. How sensitive is MCTS to the exploration coefficient c? Could the authors discuss the process of selecting this parameter and its effect?
5. Providing visual examples of successful and unsuccessful cases for DeTikZify, Claude 3, and GPT-4V would offer valuable insights into their strengths and weaknesses.
6. What types of figures or TikZ constructs does DeTikZify struggle with? What are the common failure modes of the system?

**Limitations:**

The limitations and societal impacts have been addressed.

---

> ### Author Rebuttal · Authors · 2024-08-06
>
> We appreciate the reviewer's thorough review and are pleased that our work is generally well received. We now address the remaining questions and perceived weaknesses.
>
> > The main issue that I have with this work is the potential data leakage. [...] For the public models, it is said “we only include items created after the cut-off date of CodeLLaMA and exclude repositories that may have been used in training DeepSeek.” but how to ensure this for the private models?
>
> The proprietary nature of the private models indeed makes it difficult to address data leakage and cross-contamination as we cannot make a lot of assumptions about their training data. On the upside, however, this means the performance of these private models should be interpreted as an upper bound, and since our public models manage to outperform them this strengthens our findings. We will add this to our limitations section and move it to the main text in our revised version.
>
> > From the text I find it hard to understand the intuition behind and the exact computation of “Average Similarity” (L249). Would it be possible to improve clarify this in the main text?
>
> Average scores are commonly employed to provide a holistic view of a system's performance across various metrics \[1,2]. In our case, since the metrics are on different scales, 0-1 normalization is necessary before computing their average (cf. Section 6.1). This gives a clear picture that our models dominate overall. We will elaborate on this more in the revised version.
>
> > How sensitive is MCTS to the exploration coefficient c? Could the authors discuss the process of selecting this parameter and its effect?
>
> The performance of MCTS is indeed sensitive to the exploration coefficient. A high value for c favors exploration of less-visited regions, whereas a low value emphasizes exploitation of high-value areas. Although theory provides some guidance, in practice, this parameter is empirically determined. For our work, we opted for a lower value to prioritize exploitation over exploration, due to the high computational costs of rollouts with LLMs. This consideration will be included in the revised version of our paper.
>
> > The paper mentions that SelfSim outperforms other metrics in segment-level correlation. However, the difference in Spearman's ρ appears relatively small (Table 4). Could the authors comment on the practical significance of this difference? [And] while the paper argues for the importance of segment-level performance, it's unclear how this translates to actual user experience. Can the authors elaborate on how segment-level improvements manifest in visually perceptible differences in the generated figures?
>
> Statistically, the differences in segment-level Spearman's rho in Table 4 are significant (based on a Fisher's z-Tests with a significance level of 5%). Practically, this means that when using SelfSim in MCTS, the computed rewards for rollouts better align with human perception, resulting in higher-quality outputs when exploiting high-value regions and, therefore, improved user experience (this is in contrast to system-level performance, which assesses the overall quality of whole systems and not individual outputs).
>
> > Providing visual examples of successful and unsuccessful cases for DeTikZify, Claude 3, and GPT-4V would offer valuable insights into their strengths and weaknesses.
>
> Although we already provide examples in Table 8, we agree that additional examples would offer more valuable insights, a point also raised by reviewer tgj9. Please refer to the additional examples in the supplementary pdf provided in the general response. These will be included in the revised version of our paper.
>
> > What types of figures or TikZ constructs does DeTikZify struggle with? What are the common failure modes of the system?
>
> The failure modes of DeTikZify result primarily from the composition of our dataset. For instance, humans sometimes compose independent TikZ pictures to create more complex figures. Since most examples in DaTikZ treat each TikZ picture separately, in practice it might be difficult to generate such composed figures in one go.
>
> > The paper focuses on scientific figures, which often exhibit structured layouts and simple shapes. However, TikZ's capabilities extend to much more complex diagrams and illustrations. How well would DeTikZify generalize to more diverse TikZ use cases beyond scientific figures? Are there inherent limitations in the approach that hinder its application to more free-form diagrams?
>
> As mentioned previously, the limits of DeTikZify are largely defined by our datasets, and DaTikZ contains a large portion of intricate TikZ pictures, including free-form diagrams and state machines. Some examples on how DeTikZify handles this are provided in Figure 1 and Table 8. However, even outside these bounds, we observe rudimentary emergent capabilities, for example generating TikZ code for photorealistic images. Understanding the boundaries of DeTikZify, e.g., through detailed dataset analysis and exploration of its emergent capabilities is a key aspect for our future research.
>
> \[1]: Yuan, Weizhe, Graham Neubig, and Pengfei Liu. "Bartscore: Evaluating generated text as text generation." Advances in Neural Information Processing Systems 34 (2021): 27263-27277.
>
> \[2]: Liu, Haotian, et al. "Visual instruction tuning." Advances in neural information processing systems 36 (2024).

---

> > ### Comment · Reviewer_uo3w · 2024-08-12
> >
> > I thank the authors for further clarifications.
> >
> > I still think this is a strong paper and recommend its acceptance.

---

### Official Review · Reviewer_gKyN · 2024-07-13

**Soundness:** 3
**Presentation:** 3
**Contribution:** 3
**Rating:** 7
**Confidence:** 4

**Summary:**

This paper proposes DeTikZify, a multimodal language model that generates scientific graphics in TikZ from hand-drawn illustrations. The authors provide three datasets: DaTikZ v2 (TikZ source code and figures), SketchFig (TikZ and hand-drawn sketch pairs), and SciCap++ (figures and captions). They also propose a method to iteratively improve generated TikZ using Monte Carlo Tree Search (MCTS) based on two reward functions. Experimental results show that combinations of multiple VLMs with the proposed method outperform closed-source LVMs like GPT and Claude. Additionally, iterative exploration using MCTS is shown to further improve figures efficiently.

**Strengths:**

- Tackles the challenging task of generating TikZ from hand-drawn illustrations.
- Provides three new datasets for this task.
- Experimental results sufficiently demonstrate the contribution of the proposed method. The authors conducted preliminary studies on using SVG in addition to TikZ for figure generation, and whether the evaluation metrics correlate with human subjective evaluations, before quantitatively demonstrating the effects of iterative improvement and MCTS.

**Weaknesses:**

A dataset named SciCap++ already exists and is not cited in this paper. The authors need to appropriately acknowledge this, explain the differences, and if necessary, rename their dataset.
[a] Yang et al., SciCap+: A Knowledge Augmented Dataset to Study the Challenges of Scientific Figure Captioning. AAAI workshop SDU, 2023.

**Questions:**

A response to the point raised in the Weaknesses section is expected.

**Limitations:**

As stated in the main text, the authors provide a sufficient discussion of limitations in the appendix.

---

> ### Author Rebuttal · Authors · 2024-08-06
>
> We appreciate the reviewer's time and are pleased to know they generally liked our work.
>
> > A dataset named SciCap++ already exists and is not cited in this paper.
>
> We acknowledge the potential for confusion regarding similar dataset names, despite a slight variation in the suffix (our dataset is SciCap++, while Yang et al.'s is SciCap+). However, it is important to note that these datasets serve distinct purposes: SciCap+ is derived from the original SciCap dataset and exclusively contains scientific figures of a specific type, namely graph plots. On the other hand, our meta-dataset, SciCap++, aggregates various types of scientific figures from multiple sources, including an updated version of the original SciCap dataset (which we do cite) but excluding SciCap+ due to its narrow scope. The broader range of figures in SciCap++ makes it especially well-suited for our pretraining phase aimed at generating diverse scientific figures. Additionally, our dataset is considerably larger, containing 734k figures compared to SciCap+'s 414k. In light of the reviewer's advice, we will rename our dataset in the revised version of our paper to avoid any future confusion.

---

### Official Review · Reviewer_bXux · 2024-07-17

**Soundness:** 2
**Presentation:** 2
**Contribution:** 2
**Rating:** 4
**Confidence:** 4

**Summary:**

To create high-quality scientific figures the work trains a multimodal language model, DETixZirv, a new multimodal language model from sketches and existing figures. Trained on ATIXZ.2 (over 360k human-created TikZ graphics), SKETCHFIG (hand-drawn sketches paired with scientific figures), and SciCAr++ (diverse scientific figures with metadata) datasets, MCTS-based inference algorithm to refine outputs it outperforms CLAUDE 3 and GPT-4V in creating TikZ programs.

**Strengths:**

- The work addresses a novel and innovative challenge the approach seems to be easily adapted to downstream tasks.

- The work introduces several new datasets to further scientific figures generation The work efficiently combines vision and language models for sketch to TiKz output, along with Instruct-Pix2Pix to convert instructions to draw sketch to sketch.

- Identifying the challenges pertaining to the problem statement, the work effectively dismisses certain solutions and opts for the Monte Carlo Tree Search.

- For efficient training, the work uses 2 losses one based on the success of compilation and another to measure the similarity between generated images and input images.

- To validate the performance of the model, the work proposes several metrics which though seeming trivial, fit the task well.

**Weaknesses:**

- Evaluation relies on compiler diagnostics and perceptual similarity, which may need to fully capture the quality and usability of generated TikZ code. Thus, visual outputs are necessary to evaluate the performance efficiently.

- Generated code might still contain subtle errors or inconsistencies that are not easily detectable through automated metrics.

- Though the proposed model works better than GPT-4V when run for 10 minutes, it might limit the use cases of the work especially considering the computational requirements necessary.

- From the provided metrics it could be inferred that the proposed model performs better than other models, however, the improvement provided considering the time and computational requirements makes the model less effective.

**Questions:**

Please see the weakness section.

**Limitations:**

Please see the weakness section.

---

> ### Author Rebuttal · Authors · 2024-08-06
>
> We appreciate the reviewer's time and their feedback on our work. We noted a few misunderstandings in the reviewer's summary, as well as the listed strengths and weaknesses, which we would like to clarify.
>
> First, while we generally agree with the summary, for accuracy and to prevent any confusion we want to point out that instead of DETixZirv, ATIXZ.2, and SciCAr++, our artifacts are called DeTikZify, DaTikZv2, and SciCap++.
>
> Second, although we are grateful for the recognition of our work's strengths, we must clarify that Instruct-Pix2Pix does not "convert instructions to draw sketch to sketch," but rather converts figures into sketches (according to edit instructions). Moreover, we do not use two losses for training; the described functions serve as reward functions for MCTS, which does not involve additional training. We will now address the perceived weaknesses.
>
> > Evaluation relies on compiler diagnostics and perceptual similarity [...]. Thus, visual outputs are necessary to evaluate the performance efficiently.
>
> Only the image similarity metrics depend on visual outputs (and not all of them are based solely on perceptual similarity). Code similarity and efficiency metrics do not rely on visual outputs. Thus, this statement is not fully accurate and we do not consider this a weakness. We will clarify this point in the revised version of the paper.
>
> > Generated code might still contain subtle errors or inconsistencies that are not easily detectable through automated metrics.
>
> While we acknowledge that automatic metrics are not flawless, this is a challenge common across all fields, which is why human evaluations are typically included-and we have followed this approach as well. Further, while we agree that such fidelity errors are difficult to detect in code space, they become apparent in the compiled outputs, which our image similarity metrics are designed to account for. Thus, we do not view this issue as having significant implications, and we will make this clearer in the revised version of our paper.
>
> > Though the proposed model works better than GPT-4V when run for 10 minutes, it might limit the use cases of the work especially considering the computational requirements necessary.
>
> We agree that there is a quality versus run-time trade-off in time-budgeted versus output-oriented inference. However, this applies not only to our models but also to the baselines, including GPT-4V; and our results indicate that our models are competitive or outperform the baselines given equivalent inference methods (i.e., comparable run-time). Also, even when comparing output-oriented GPT-4V with time-budgeted DeTikZify across inference modes, as the reviewer suggests, given GPT-4V's presumably much larger size, DeTikZify still likely requires much fewer computational resources.
>
> > From the provided metrics it could be inferred that the proposed model performs better than other models, however, the improvement provided considering the time and computational requirements makes the model less effective.
>
> As mentioned in the previous point, this trade-off is not unique to our models but applies to all evaluated models. Given this context, we kindly request the reviewer to reconsider their evaluation of our work.

---

### Author Rebuttal · Authors · 2024-08-07

We would like to extend our gratitude to all the reviewers for their feedback and we are delighted that our work was generally well-received. In response to the comments from reviewers uo3w and tgj9, we have uploaded a PDF containing additional examples along with this general response. For more details, see our individual responses below.

---

### Decision · Program_Chairs · 2024-09-25

**Decision:**

Accept (spotlight)

**Comment:**

Three of the four reviewers gave either a strong accept or accept for this paper.  Those reviewers appreciated the novelty of the work to automatically generate TikZ code from both existing figures and hand-drawn sketches, and they felt the results and evaluations were conceiving.

There was one dissenting reviewer.  This review had some inconsistencies in their comments that don't quite line up with the paper or are unclear and the reviewer didn't engage in the discussion with the authors or other reviewers.  They noted there were no visual results, however those appear in the appendix.

Given the discussion with the other reviewers and the consideration of the rebuttal which other reviewers did respond to and felt satisfied, the AC agrees that this paper is a strong contribution and is accepted.

Congratulations!

In revising your paper for the camera-ready, please include and comments and clarifications that were made or proposed during the rebuttal and discussion phase.  Also, the AC recommends including some visual results that are in the appendix into the main body of the paper.